# Activation of CAMK2 by pseudokinase PEAK1 represents a targetable pathway in triple negative breast cancer

Xue Yang [1,2], Xiuquan Ma [1,2], Tianyue Zhao[1,2], David R. Croucher [3,4], Elizabeth V. Nguyen[1,2], Kimberley C. Clark [1,2], Changyuan Hu [1,2], Sharissa L. Latham [3,4], Charles Bayly-Jones [1,2], Bao V. Nguyen [5], Srikanth Budnar [1,2], Sung-Young Shin [1,2], Lan K. Nguyen [1,2,6], Thomas R. Cotton [1,2], Anderly C. Chüeh [1,2], Terry C. C. Lim Kam Sian [1,2], Margaret M. Stratton [5], Andrew M. Ellisdon [1,2] & Roger J. Daly [1,2] ✉

The PEAK family of pseudokinases, comprising PEAK1-3, play oncogenic roles in several poor prognosis human cancers, including triple negative breast cancer (TNBC). However, therapeutic targeting of pseudokinases is challenging due to their lack of catalytic activity. To address this, we screen for PEAK1 effectors and identify calcium/calmodulin-dependent protein kinase 2 (CAMK2)D and CAMK2G. PEAK1 promotes CAMK2 activation in TNBC cells via PLCγ1/Ca$^{2+}$ signalling and direct binding to CAMK2. In turn, CAMK2 phosphorylates PEAK1 to enhance association with PEAK2, which is critical for PEAK1 oncogenic signalling. To achieve pharmacologic targeting of PEAK1/ CAMK2, we repurpose RA306, a second generation CAMK2 inhibitor. RA306 inhibits PEAK1-enhanced migration and invasion of TNBC cells in vitro and significantly attenuates TNBC xenograft growth and metastasis in a manner mirrored by PEAK1 ablation. Overall, these studies establish PEAK1 as a critical cell signalling nexus that integrates Ca$^{2+}$ and tyrosine kinase signals and identify CAMK2 as a therapeutically 'actionable' target downstream of PEAK1.

Pseudokinases play critical roles in normal cellular and developmental processes as well as in human diseases such as cancer, despite lacking the kinase activity of their bona fide kinase counterparts. This reflects the ability of pseudokinases to function as allosteric regulators and/or scaffolds that assemble protein signalling complexes[1,2]. Due to their pathological functions, pseudokinases are attracting increasing attention as therapeutic targets[3].

The PEAK family of pseudokinases, comprising PEAK1 (previously known as SgK269), PEAK2 (Pragmin and SgK223), and PEAK3 (C19orf35) share a conserved molecular architecture and sequence homology, and function as scaffolds to mediate and regulate signalling

downstream of particular growth factor receptors and integrins[4,5]. PEAK1 is a critical regulator of EGFR signalling, mediating a qualitative shift in signal output over time from mitogenic/survival signalling to that promoting cell migration/invasion[16]. In addition, PEAK1 is strongly implicated in the progression of a variety of cancers. For example, in breast cancer, PEAK1 promotes cell proliferation, epithelial-mesenchymal transition (EMT), aberrant acinar morphology in 3D culture and xenograft growth and metastasis[6–8], in non-small cell lung cancer it enhances migration, invasion, EMT and experimental metastasis[9], while in pancreatic cancer models it promotes resistance to specific therapies[10]. Of particular interest, high PEAK1 expression

[1]Cancer Program, Monash Biomedicine Discovery Institute, Clayton, VIC, Australia. [2]Department of Biochemistry and Molecular Biology, Monash University, Melbourne, VIC, Australia. [3]Garvan Institute of Medical Research, Darlinghurst, NSW, Australia. [4]St Vincent's Clinical School, Faculty of Medicine, UNSW Sydney, Darlinghurst, NSW, Australia. [5]Department of Biochemistry and Molecular Biology, University of Massachusetts, Amherst, MA, USA. [6]Present address: South Australian Immunogenomics Cancer Institute, University of Adelaide, Adelaide, Australia. ✉e-mail: roger.daly@monash.edu

occurs in triple negative breast cancer (TNBC) and pancreatic cancer[6,10], two cancers associated with poor prognosis and limited targeted therapeutic options. PEAK2 and PEAK3 also enhance cell migration and invasion and are overexpressed in specific malignancies[4,11-15].

All PEAK family members exhibit a C-terminal pseudokinase domain flanked by a split helical dimerisation (SHED) domain, and N-terminal extensions of contrasting length that harbour recruitment sites for specific SH2, SH3 and PTB domain-containing effectors. These effectors include Grb2 (bound by PEAK1 and PEAK3), Shc1 (PEAK1), C-terminal Src kinase (Csk) (PEAK2) and ASAP1, Cbl and PYK2 (PEAK3)[4,5]. An important characteristic of the PEAK family is their ability to form homotypic and heterotypic complexes[4,16]. The dimerisation of the PEAKs is mediated by the α-helical SHED domain that forms a highly conserved dimer interface driven by hydrophobic interactions[17,18], while higher-order oligomers can also form via pseudokinase domain interfaces[18]. Dimerisation is critical to the signalling potential of PEAK scaffolds[4,12,18]. Furthermore, since the repertoire of known interactors differs across this family of scaffolds, heterotypic versus homotypic association acts to diversify signal output[4,16]. For example, PEAK1 requires PEAK2 to efficiently activate Stat3 and promote migration in MCF-10A cells[16,19], while signalling by PEAK3 is modulated both quantitatively and temporally by the presence of PEAK1 and PEAK2[4].

In this study, we address how homo- and heterotypic association of PEAK1 and PEAK2 regulates signal output by defining the interactomes of the corresponding dimeric complexes. We identify the association of PEAK1 with calcium/calmodulin-dependent protein kinase 2 (CAMK2)D and CAMK2G, of particular interest given the known roles of specific CAMK2 isoforms in many human cancers, including breast cancer, and the therapeutic 'actionability' of these serine/threonine kinases[20,21]. Importantly, we demonstrate potent, feed-forward activation of CAMK2 by PEAK1, highlighting PEAK1 as a critical regulator and integrator of tyrosine kinase and $Ca^{2+}$ signals. In addition, we identify CAMK2D and CAMK2G as downstream effectors of PEAK1 in TNBC and 'repurposing' of the second generation CAMK2 inhibitor RA306 as a potential therapeutic strategy against oncogenic PEAK1 signalling in poor prognosis human cancers such as TNBC.

## Results

### Characterisation of dimer-specific interactomes of PEAK1 and PEAK2 by BiCAP

To characterise the interactomes of PEAK1 and PEAK2 homodimers, and the PEAK1/PEAK2 heterodimer, we applied bimolecular complementation affinity purification coupled with tandem mass spectrometry (BiCAP-MS/MS) (Supplementary Fig 1A)[22]. Specific dimeric complexes were affinity purified, and associated proteins identified by LC-MS/MS in data-independent acquisition mode (Methods, Fig. 1A–C and Supplementary Table 1 in Source Data). Strong correlation between $n = 3$ independent experiments was observed (mean sample correlation scores: Vector = 0.78; PEAK1 homodimer = 0.85; PEAK2 homodimer = 0.94; PEAK1/2 heterodimer = 0.94) (Supplementary Fig 1B). Grb2, a well-characterised PEAK1 binding partner[6,16] bound the PEAK1 homodimer and PEAK1/PEAK2 heterodimer, but not the PEAK2 homodimer, consistent with our previous results that PEAK1 bridges Grb2 to PEAK2[16]. Two protein phosphatase 1 (PP1) family members, PPP1CA and PPP1CC were previously identified as PEAK1 interactors in a traditional immunoaffinity based-MS/MS screen[19]. However, their selectivity towards PEAK1 versus PEAK2 was unclear. Here, like Grb2, they were identified as interactors of the PEAK1 homodimer and PEAK1/PEAK2 heterodimer, with another PP1 family member, PPP1CB, associating with the PEAK1 homodimer. Of interest given a recent paper reporting a key regulatory role for 14-3-3 proteins in regulating PEAK3 effector recruitment and biological activity[23], multiple 14-3-3 isoforms bound both homodimers and the

heterodimer (Supplementary Table 1 in Source Data). Further interactors identified included the RAC1 guanine nucleotide exchange factor (GEF) FARP1 and the MAPK pathway antagonist SPRY4, which were selective for the PEAK2 homodimer. In addition, $Ca^{2+}$/calmodulin-dependent protein kinase 2 (CAMK2)D and G associated with the PEAK1 and PEAK2 homodimers and the PEAK1/PEAK2 heterodimer, while CAMK2B was recruited by the PEAK2 homodimer and PEAK1/PEAK2 heterodimer (Fig. 1A–C). Bioinformatic analysis of the interactomes for each of the PEAK complexes in order to identify enriched pathways/processes identified calcium-related pathways that included CAMK2 enzymes as being enriched (specifically 'postsynaptic signalling' for PEAK1 and PEAK2 homotypic complexes and both 'myometrial' and 'cardiac signalling' for all 3 complexes) (Supplementary Table 2 in Source Data).

Overall, these data provided important insights into the binding selectivity of PEAK1/2 complexes and identified specific CAMK2 family members as interactors of these pseudokinase scaffolds.

### Characterisation of CAMK2 interaction with PEAK1 and PEAK2
To confirm the association of specific CAMK2 family members with PEAK1 and PEAK2, we focused on CAMK2D and CAMK2G since these kinases are associated with all of the homo- and heterotypic PEAK complexes analysed by BiCAP-MS/MS. The interaction of these CAMK2s with endogenous PEAK1 and PEAK2 was interrogated by proximity ligation assays in MDA-MB-231 breast cancer cells (Fig. 1D–F). Using this technique, the interactions between PEAK1 and CAMK2D or G could be detected and were significantly decreased when PEAK1 was knocked down using siRNA (Fig. 1D, E). Similar results were observed for the interactions between PEAK2 and CAMK2D or G (Fig. 1D, F). These data demonstrate that CAMK2D and G interact with both PEAK1 and PEAK2 in vivo, at endogenous expression levels.

### Regulation of CAMK2 activity by PEAK1
The known overexpression of PEAK1 in breast cancer[6,7] led us to interrogate the functional consequences of PEAK1/CAMK2 interaction. Upon an increase in intracellular $Ca^{2+}$, binding of $Ca^{2+}$/calmodulin (CaM) to the CAMK2 regulatory segment leads to release of the regulatory segment from the kinase domain, kinase activation, and phosphorylation of T286 (T287 in CAMK2D and G). In the presence of T286 phosphorylation, the CAMK2 regulatory segment can no longer mediate autoinhibition, and the activity of CAMK2 is $Ca^{2+}$/CaM autonomous[24]. Knockdown of PEAK1 in MDA-MB-231 TNBC cells did not affect the expression levels of CAMK2D and G but significantly reduced their activation, as determined by phosphorylation at T287 (Fig. 2A, B). In addition, overexpression of PEAK1 in these cells significantly enhanced the activation of these two CAMK2 isoforms (Fig. 2C, D). Positive regulation of CAMK2D/G activity by PEAK1 was also observed in MDA-MB-468 TNBC cells (Supplementary Fig. 2). Treatment of MDA-MB-231 cells with BAPTA-AM, a membrane permeable $Ca^{2+}$ chelator which blocks activation of CaM by $Ca^{2+}$ [25] significantly reduced basal CAMK2 T287 phosphorylation and completely blocked the effect of PEAK1 on CAMK2 activation, indicating that PEAK1-mediated CAMK2 activation is $Ca^{2+}$-dependent (Fig. 2E, F).

In light of this result, we determined whether PEAK1 can modulate $Ca^{2+}$ levels and thereby indirectly regulate CAMK2 activation. Application of Fluo-4 imaging revealed that PEAK1 overexpression significantly enhanced intracellular $Ca^{2+}$ levels at steady state (Fig. 2G, H). With regard to the mechanism underpinning PEAK1-regulated $Ca^{2+}$ signalling, the known roles of the PEAKs as regulators and/or substrates of specific tyrosine kinases[4,5] led us to determine whether PEAK1 overexpression impacts tyrosine phosphorylation and hence activation of PLCγ1, leading to generation of $IP_3$ that triggers $Ca^{2+}$ release from intracellular stores[26]. Indeed, PEAK1 knockdown decreased tyrosine phosphorylation of PLCγ1 Y783 (Fig. 2I), while PEAK1 overexpression enhanced PLCγ1 tyrosine phosphorylation on

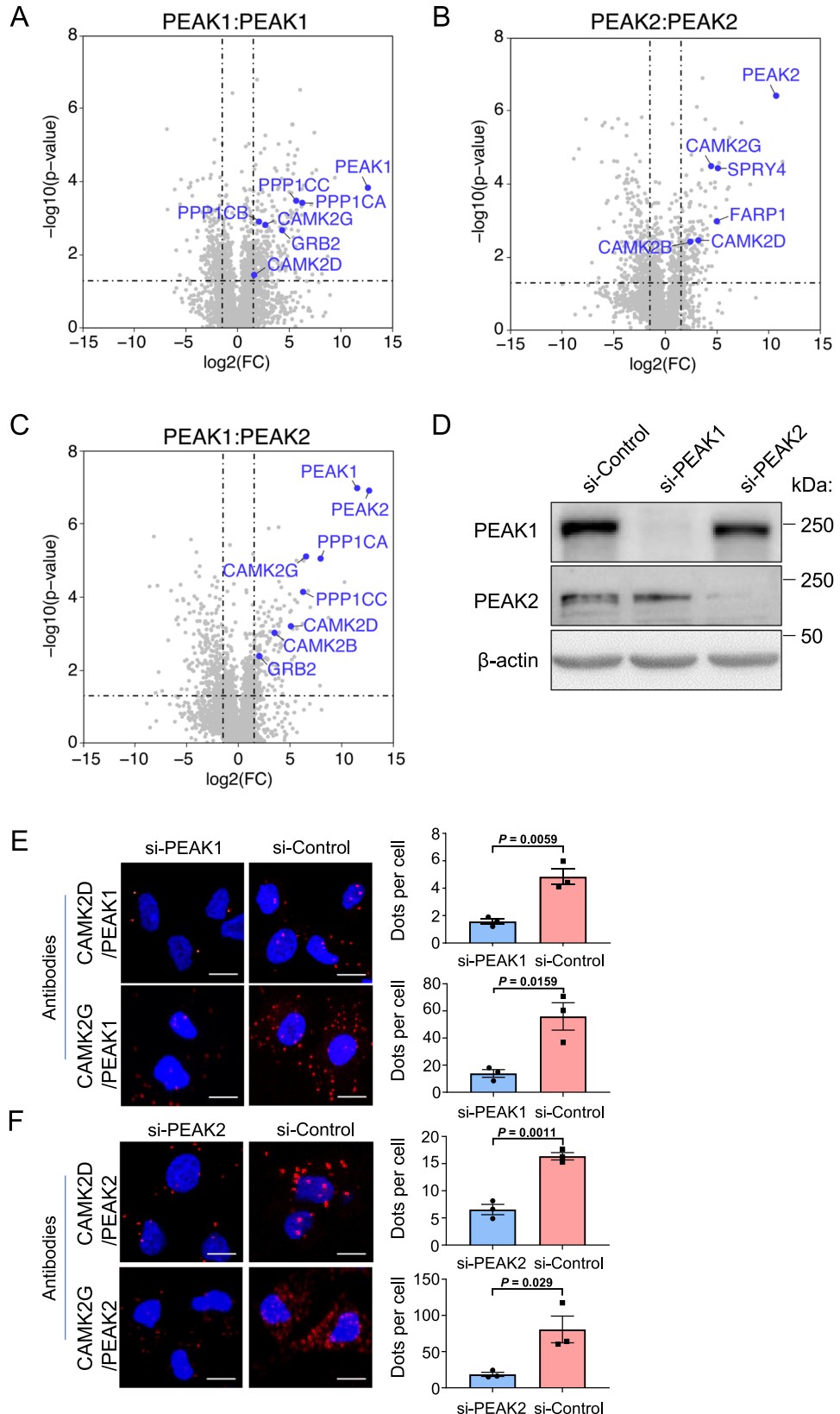

this site (Fig. 2J). However, given that PEAK1 is a pseudokinase, PLCγ1 phosphorylation must be mediated by a bona fide tyrosine kinase acting downstream of PEAK1. Since PEAK1 is a member of a prominent Src family kinase (SFK) signalling network in TNBC[27] and PLCγ1 is a known substrate of SFKs[28], we hypothesised that SFKs may mediate this role. Supporting this hypothesis, treatment of MDA-MB-231 TNBC

cells with the highly selective SFK inhibitor eCF506[29] blocked PEAK1-enhanced PLCγ1 tyrosine phosphorylation (Supplementary Fig. 3). However, interrogation of the BiCAP datasets did not reveal the association of specific SFKs or PLCγ1 with any of the PEAK complexes, strongly suggesting that the regulatory mechanism does not involve direct binding.

**Fig. 1 | Identification of specific CAMK2 isoforms as PEAK1/2 interactors via BiCAP-MS/MS. A–C.** Results of BiCAP-MS/MS screen. Volcano plots indicate enriched interactors for the PEAK1 homotypic complex (**A**), PEAK2 homotypic complex (**B**) and PEAK1/PEAK2 heterotypic complex (**C**). The known PEAK1 interactor Grb2 and targets of particular interest, including specific CAMK2 family members, are highlighted in blue. Dotted lines indicate fold change and $p$-value cut-offs. Data are from $n = 3$ independent experiments, and statistical significance is determined by a two-tailed $t$ test. **D** Western blotting validation of PEAK1 and PEAK2 knockdown. PEAK1 and PEAK2 were knocked down using siRNA pools, with successful knockdown confirmed by Western blotting. Size markers (in kDa) are indicated on the right. These are representative data from $n = 3$ independent experiments detailed below. **E, F.** Validation of PEAK1/2 association with CAMK2D and CAMK2G by proximity ligation assays. Left panels, representative proximity ligation assay (PLA) images from MDA-MB-231 cells for the interaction of PEAK1 (**E**) or PEAK2 (**F**) with CAMK2D and CAMK2G. Cells were transfected with either si-PEAK1 (**E**), si-PEAK2 (**F**) or si-Control. Scale bars indicate 10 μm. Right panels, quantification of protein interactions in each cell. The data are expressed as dot numbers divided by cell number. In each condition, more than 50 cells have been counted. Data are presented as mean values +/− standard error of the mean (SEM) from $n = 3$ independent experiments and analysed by unpaired two-tailed $t$ test. Source data are provided as a Source Data file.

## Structural requirements for PEAK1/2 interaction with CAMK2

To determine which regions of PEAK1 and PEAK2 mediate interaction with CAMK2D and G, we utilised previously described deletion mutants of PEAK1 and PEAK2[16] that delete the N-terminal region (ΔNterm), N1 α-helix (ΔN1) and a combination of the pseudokinase domain and helices J-M (ΔCterm), as well as a mutant comprising just the extreme N-terminal region (1-324 amino acids) (N-term) (Fig. 3A). Co-immunoprecipitation of Flag-tagged PEAK1 with endogenous CAMK2D could be demonstrated for WT, ΔN1 and ΔCterm PEAK1 but not for the ΔNterm PEAK1 mutant (Fig. 3B). However, the Nterm region alone did not associate with CAMK2D, indicating that it is required but not sufficient for interaction with these CAMK2s (Fig. 3B). In addition, since the N1 and Cterm regions are required for PEAK1 to undergo homotypic association and also heterotypic interaction with PEAK2[16], these data also indicated that dimerisation is not required for PEAK1 to associate with these CAMK2s. Next, we generated a series of deletion mutants to more narrowly define the CAMK2 binding region on PEAK1 (Fig. 3C). Association with CAMK2D was lost with the Δ261−324 mutant (Fig. 3D). Furthermore, deletion of smaller regions spanning amino acids 261−300 or 301−324 also abolished CAMK2D association (Fig. 3D). Deletion of amino acids 301−324 also markedly reduced CAMK2G association (Fig. 3E).

Although PEAK1 and PEAK2 share similar overall domain architecture, the N-terminal extensions, including the region of PEAK1 required for CAMK2 association (amino acids 261−324), are poorly conserved[16], indicating that the CAMK2 binding mechanism for PEAK2 is likely to differ. Consistent with this hypothesis, while CAMK2D association with PEAK2 was significantly reduced with the PEAK2 ΔNterm mutant, the greatest effect was observed with the ΔN1 and ΔCterm mutants, which markedly and almost completely abolished association, respectively (Supplementary Fig. 4A−C). Since the N1 and Cterm regions are required for PEAK2 to undergo homotypic and heterotypic association[16], these data suggest that PEAK2 interaction with these CAMK2s requires PEAK2 dimerisation. Supporting this model, three point mutations in PEAK2 (Supplementary Fig 4A) that abrogate dimerisation[18] all led to a diminished association with CAMK2D (Supplementary Fig 4D). This contrasts with PEAK1, where the dimerisation-impaired ΔN1 and ΔCterm mutants retained association with the CAMK2s. In addition, PEAK1 knockdown did not disrupt the association of PEAK2 with CAMK2D (Supplementary Fig. 4E), indicating that PEAK2 interaction with CAMK2D is not mediated via PEAK1, consistent with the BiCAP data (Fig. 1B). Taken together, these data highlight contrasting structural requirements for PEAK1 and PEAK2 association with CAMK2, with a greater contribution from a discrete N-terminal region in the case of PEAK1, and a requirement for dimerisation in the case of PEAK2.

## PEAK1 directly binds CAMK2 via a conserved motif to promote CAMK2 activation and PEAK1 phosphorylation

Since the PEAK1 Δ301−324 mutant was deficient in CAMK2 association (Fig. 3D, E), we determined the effect of this mutant on CAMK2 activation. Interestingly, this mutant was significantly impaired in its ability to activate CAMK2D/G as determined by phosphorylation at T287 (Fig. 3F, G), indicating that PEAK1-mediated CAMK2 activation requires association of the two proteins in addition to a $Ca^{2+}$ signal (Fig. 2E, F).

A recent report characterised how CAMK2 interacts with substrate binding partners via a conserved sequence motif in the binding partner, R/K-X-X-ϕ-X-R/K-X-X-S/T-ϕ (where ϕ represents a hydrophobic residue) that binds across a single continuous site on the CAMK2 kinase domain[30]. It also revealed that binding of the conserved CAMK2 interaction motif (CIM) competes with autoinhibition of CAMK2 by the regulatory segment of the kinase domain, explaining the ability of certain substrates, such as GluN2B, to lock CAMK2 in an active conformation[30]. Of note, the Rac GDP/GTP exchange factor TIAM1 exhibits a pseudosubstrate version of the conserved motif (with A rather than S/T) but is still able to bind and promote phosphorylation of itself on other sites[31]. To address whether PEAK1 may interact with CAMK2 via this mechanism, we searched the PEAK1 N-terminal region for the conserved motif. We first constructed a multiple sequence alignment across diverse species and subsequently subjected this to MEME analysis to identify conserved motifs. This analysis revealed that the PEAK1 amino acids 297−307 exhibits close similarity to the canonical binding motif but with an alanine at position 0 (amino acid 306), as observed with TIAM1 (Fig. 4A). Furthermore, the N-terminal region and the binding motif are highly conserved across species (Supplementary Fig. 5).

The presence of a CIM between amino acids 297−307 is consistent with disruption of binding by deletion of amino acids 261−300 and 301−324 (Fig. 3D, E) and the failure of the Δ301−324 mutant to activate CAMK2 (Fig. 3F, G). To determine whether a synthetic peptide corresponding to the PEAK1 CIM bound CAMK2, we applied isothermal calorimetry. These data revealed a high-affinity interaction with a mean dissociation constant ($K_D$) of 635 nM (Fig. 4B), comparable to that of the TIAM1 interacting region (1.1 μM)[30]. Modelling of the PEAK1/CAMK2 interaction using AlphaFold predicts (pLDDT 90.7, pTM 0.899) that the PEAK1 CIM interacts across the same surface of the CAMK2 kinase domain as other known interactors[30], with key salt bridges formed between PEAK1 R303 (-3 position) and R297 (-9) with glutamate residues in CAMK2 (Fig. 4C−F and Supplementary Fig 6). In addition, L301 is accommodated in a hydrophobic pocket of CAMK2 (Fig. 4E). We also note the predicted formation of salt bridges and electrostatic interactions by PEAK1 D308, D310, D311 and D314 across the surface of CAMK2 (Fig. 4F). Of note, while this modelling was undertaken with CAMK2A, the binding site is conserved across all CAMK2 isoforms, as highlighted by AlphaFold2 modelling and multiple sequence alignments of CAMK2A, B, D and G (Supplementary Fig. 6).

To test the importance of key residues within the PEAK1 CIM for CAMK2 activation, we generated PEAK1 mutants with alanine substitutions at the R303 (-3), L301 (-5) and R297 (-9) positions (termed AAA) and glutamate substitutions at R303 and R297 (EE). Over-expression of the EE mutant significantly enhanced intracellular $Ca^{2+}$ levels (Fig. 5A, B) and both mutants significantly increased PLCγ1 tyrosine phosphorylation (Fig. 5C, D). However, both mutants lost the ability to activate CAMK2 as determined by phosphorylation at T287 (Fig. 5E−G). These data indicate that while the ability of PEAK1 to activate CAMK2 is calcium-dependent (Fig. 2E, F), PEAK1-enhanced

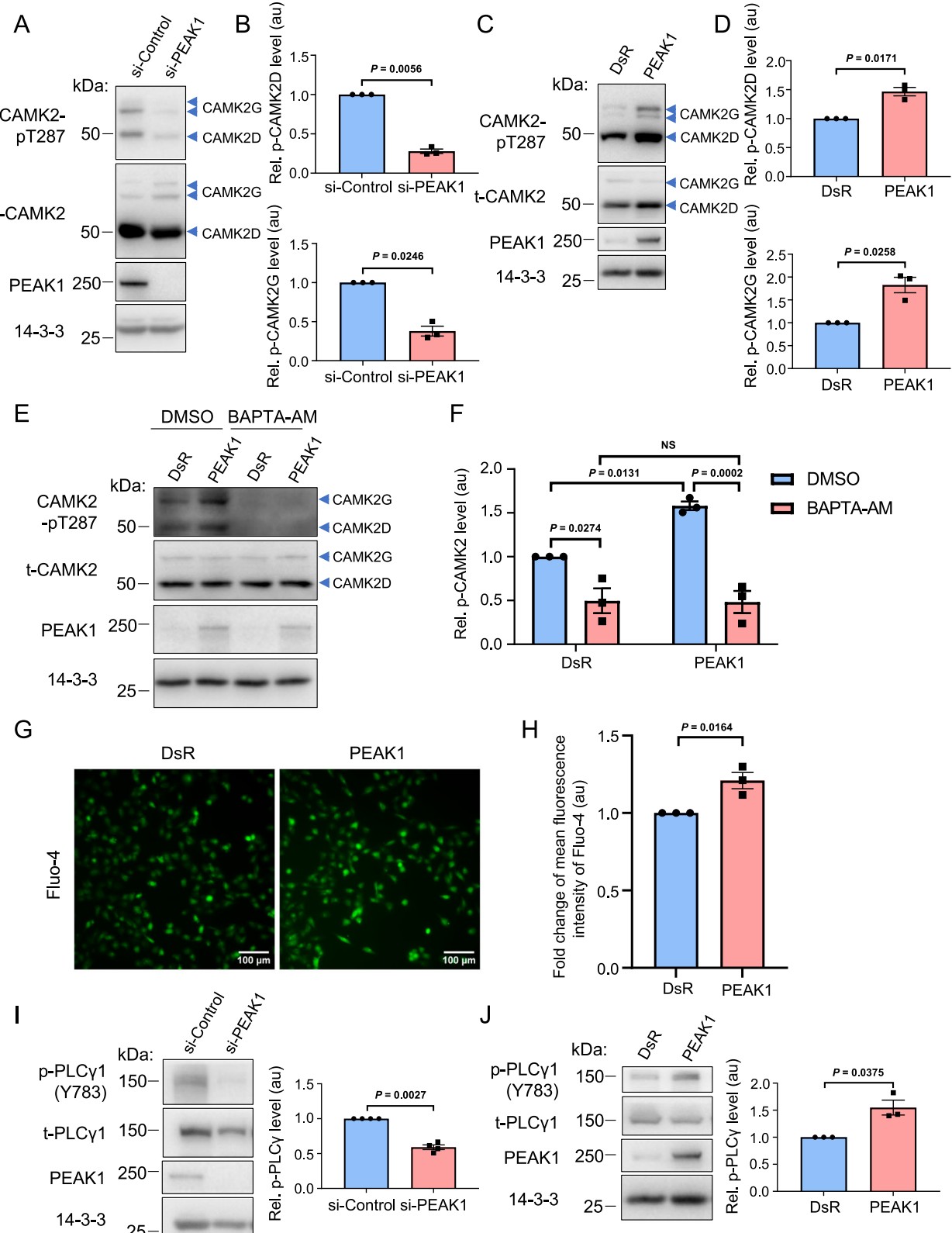

Ca²⁺ is not sufficient to increase CAMK2 T287 phosphorylation, and direct association with CAMK2 is also required. The data also support the validity of the Alphafold modelling.

CAMK2 binding to TIAM1 leads to sustained phosphorylation of TIAM1[31]. To determine whether PEAK1 is a substrate for CAMK2, we utilised RA306, a 'next generation' CAMK2 inhibitor developed by Sanofi for chronic cardiovascular indications that is selective for CAMK2D and G over other isoforms[32]. RA306 exhibited 'on-target' activity against CAMK2 in MDA-MB-231 cells (Supplementary Fig. 7A) and significantly inhibited PEAK1 serine/threonine phosphorylation (Fig. 6A, B). In addition, recombinant CAMK2 demonstrated in vitro kinase activity towards PEAK1 that was detectable using a pAkt substrate antibody that also recognises the minimal CAMK2 consensus motif for phosphorylation (RXXS/T) or a pSer/Thr antibody (Fig. 6C).

**Fig. 2 | PEAK1-mediated regulation of CAMK2 activity in triple negative breast cancer cells. A–D** Manipulation of PEAK1 expression modulates CAMK2 activity. PEAK1 was knocked down using a siRNA (**A**, **B**) or overexpressed via transient transfection (**C**, **D**) in MDA-MB-231 cells and autonomous activation of CAMK2 determined by Western blotting as indicated. Blotting for 14-3-3 served as a loading control. Histograms indicate CAMK2 T287 phosphorylation normalised for total CAMK2 expression, with 'au' indicating arbitrary units. **E**, **F** PEAK1-promoted CAMK2 activation is calcium-dependent. MDA-MB-231 cells were transfected with a PEAK1 construct in the presence or absence of 10 μM BAPTA-AM. Cell lysates were Western blotted as indicated (**E**), and CAMK2 activation quantified as above for CAMK2D/G combined (**F**), with data expressed relative to the DsR (vector)/DMSO control which was arbitrarily set at 1. **G**, **H**. Overexpression of PEAK1 increases intracellular Ca²⁺. MDA-MB-231_EcoR cells stably overexpressing PEAK1 were loaded with calcium-sensitive dye Fluo-4 AM and then imaged with a fluorescence microscope. Representative images from $n = 3$ independent experiments are shown (**G**) as well as quantification of Fluo-4 images (**H**), presented as mean Fluo-4 fluorescence ± SEM. **I**, **J** PEAK1 regulates tyrosine phosphorylation of PLCγ1. MDA-MB-231 cells were transfected with PEAK1 siRNA (**I**) or a PEAK1 expression vector (**J**) and then cell lysates were Western blotted as indicated. The histograms indicate Y783-phosphorylated PLCγ1 normalised for total PLCγ1 expression. Data are presented as mean values +/− SEM. For (**B**, **D**, **F**, **H** and **J**), data are from $n = 3$ independent experiments, for **I**, $n = 4$ independent experiments. NS indicates $p > 0.05$. Data were analysed by ratio paired two-tailed $t$ test (**B**, **D**, **I**, **J**), two-way ANOVA with Tukey's multiple comparisons test (**F**), or unpaired two-tailed $t$ test (**H**). PEAK1 and 14-3-3 blots in (**C** and **J**) are from the same experiment. Source data are provided as a Source Data file.

Interestingly, PEAK1 phosphorylation was markedly reduced when the PEAK1 AAA and EE CIM mutants were used as substrate, indicating that efficient phosphorylation of PEAK1 by CAMK2 requires interaction of the two proteins via the PEAK1 CIM (Fig. 6C).

## CAMK2 phosphorylates PEAK1 to regulate the PEAK1 interactome

In order to characterise the impact of CAMK2-mediated phosphorylation on PEAK1 scaffolding potential, we compared the interactomes of wild-type (WT) PEAK1 and the AAA and EE CIM mutants in vivo. We utilised this approach since it more selectively addresses the impact of CAMK2-mediated PEAK1 phosphorylation, while RA306 treatment would exert a more general inhibition of CAMK2-mediated cellular phosphorylation. The binding of the adaptors Crk-L and Grb2, which associate with PEAK1 via SH3 and SH2 domain-mediated interactions, respectively[9,23], was unaffected by the CIM mutations (Supplementary Fig. 7B). In contrast, Western blotting of PEAK1 IPs revealed significantly reduced association of 14-3-3 proteins with the two mutants (Supplementary Fig. 7C, D). Since 14-3-3 binding to client proteins is classically phosphorylation-dependent, this indicated that impaired CAMK2 activation by the CIM mutants leads to decreased phosphorylation of PEAK1 on 14-3-3 binding sites. Next, we assayed the heterotypic association of PEAK1 with PEAK2, a critical determinant of signal output, with PEAK2 binding being required for PEAK1 to promote cell migration and oncogenic signalling[16]. Importantly, whereas the heterotypic association of wildtype PEAK1 with PEAK2 in MDA-MB-231 TNBC cells could be readily detected, this was markedly reduced for the CIM mutants (Fig. 6D). Overall, these data demonstrate that CAMK2-mediated PEAK1 phosphorylation is a critical regulator of PEAK1-mediated protein-protein interactions and signalling.

## PEAK1/CAMK2 signalling is a therapeutically 'actionable' target in TNBC

The finding that CAMK2 and PEAK1 reciprocally regulate led us to interrogate how the expression of these proteins is related to breast cancer patient survival. Interestingly, combined high expression of PEAK1 and CAMK2D was significantly associated with poor overall survival, while a non-significant trend was detected for the PEAK1/CAMK2G combination (Supplementary Fig. 8A–C). Of note, this analysis emphasised the importance of high expression of both PEAK1 and CAMK2D (Supplementary Fig. 8B), consistent with a co-activating PEAK1/CAMK2D 'cassette' promoting breast cancer progression. To gain further insights into the combined role of PEAK1 and CAMK2/G in breast cancer, we interrogated their association with distant metastasis-free survival (DMFS). However, this was only possible with a meta-cohort derived from certain publicly available datasets (see Methods) and for the PEAK1/CAMK2G combination. Nevertheless, this revealed a significant association between high PEAK1/CAMK2G and reduced DMFS (Supplementary Fig. 8D).

Next, we characterised the effects of PEAK1/CAMK2 signalling on biological endpoints in TNBC cells. Previously we reported that

CRISPR-mediated PEAK1 KO in MDA-MB-231 cells significantly reduced anchorage-dependent proliferation as well as tumoursphere formation in low attachment plates[8]. Interestingly, treatment with RA306 resulted in a dose-dependent reduction in the viability of these cells under anchorage-dependent and -independent conditions (high and low attachment plates, respectively) (Supplementary Fig. 9A, B). These effects appeared to be mediated via decreased proliferation, as PEAK1 KO or RA306 did not induce apoptosis (Supplementary Fig. 9C, D). Taken together, these data indicate that both PEAK1 and CAMK2 contribute to proliferative signalling in TNBC.

Next, we probed the role of PEAK1/CAMK2 in cell migration/invasion. Knockdown of either CAMK2D or G blocked the effect of PEAK1 overexpression on cell migration and invasion (Fig. 7A, B and Supplementary Fig. 10A) and RA306 treatment blocked PEAK1-promoted migration in both MDA-MB-231 and -468 cells (Fig. 7C and Supplementary Fig. 10B, C) and PEAK1-promoted invasion in MDA-MB-231 cells (Supplementary Fig. 10D). Furthermore, the Δ301−324 PEAK1 mutant, which cannot bind or activate CAMK2 (Fig. 3D−G), was unable to enhance TNBC cell migration and invasion (Fig. 7D and Supplementary Fig. 10E). Overall, these data indicate a requirement for CAMK2 expression and/or activation, as well as PEAK1/CAMK2 binding, in regulation of cell migration and invasion by PEAK1.

We then evaluated RA306 in an in vivo model of TNBC tumour growth and metastasis. Previously, we demonstrated that PEAK1 gene knock-out significantly inhibited growth and lung metastasis of orthotopic MDA-MB-231 TNBC xenografts[8]. To test RA306, we utilised the highly metastatic variant of this cell line, MDA-MB-231-HM[33], so that we could readily assay metastatic spread from the primary tumour. In addition, we generated PEAK1-deficient MDA-MB-231-HM cells via CRISPR editing (Supplementary Fig. 11A) to compare the effect of RA306 with PEAK1 knock-out and also the combined effect of these two manipulations. On-target activity of RA306 could be detected in the xenografts, and PEAK1 ablation also reduced CAMK2 activation (Supplementary Fig 11B, C). RA306 treatment significantly impaired primary tumour growth (Fig. 7E). Importantly, the degree of growth inhibition was similar to that observed upon PEAK1 gene knock-out and combining RA306 with gene knock-out did not enhance this further, strongly suggesting that RA306 treatment and PEAK1 knock-out are targeting the same oncogenic signalling axis (Fig. 7E). Furthermore, we also allowed a separate cohort of primary tumours to grow to the same mean size (Supplementary Fig. 11D), resected them and then monitored metastasis. Strikingly, RA306 blocked metastasis to a similar extent to PEAK1 knock-out (Fig. 7F and Supplementary Fig. 11E). Since PEAK1 KO or RA306 treatment impacts cell proliferation, migration and invasion but not apoptosis[8] (Fig. 7 and Supplementary Figs. 9, 10), the effect on primary tumour growth is likely mediated via reduced proliferation while the reduced metastatic growth must be independent of primary tumour size and reflect a decreased ability to metastasise to the lung and/or reduced proliferation at this secondary site. Overall, these data identify RA306 as a potential therapeutic strategy against the PEAK1/CAMK2 axis in TNBC and potentially other human cancers.

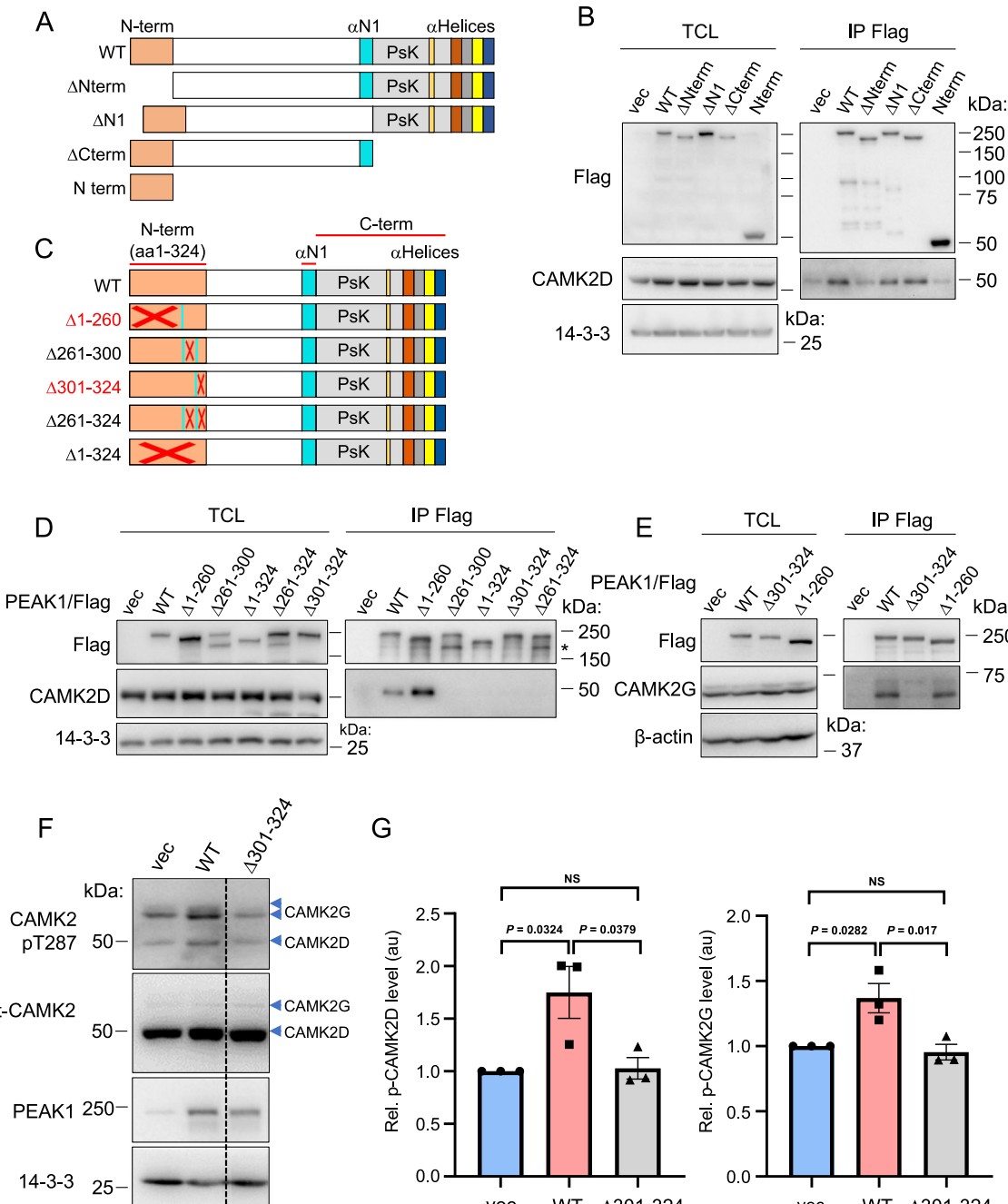

**Fig. 3 | Mapping the CAMK2 binding region on PEAK1. A** Schematic structure of PEAK1 and various truncation mutants. WT, wildtype. **B** Association of CAMK2D with specific PEAK1 deletion mutants. The indicated Flag-tagged PEAK1 deletion mutants were expressed in HEK293T cells. Total cell lysates (TCL) and Flag IPs were subjected to Western blotting as indicated. Positions of size markers are indicated for both panels, sizes on the right. **C** Schematic representation of various PEAK1 N-terminal deletion mutants. Deleted regions of PEAK1 are indicated by red crosses. **D**, **E**. Fine structure mapping of N-terminal regions required for the association of PEAK1 with CAMK2D (D) or CAMK2G (E). Analysis was undertaken as in (**B**). The asterisk in 3D highlights degradation products present for the Δ261−300 and Δ261−324 mutants. **F**, **G**. PEAK1 binding to CAMK2 is required for CAMK2 activation.

MDA-MB-231 cells were transfected with WT PEAK1 or the Δ301−324 PEAK1 construct. Cell lysates were Western blotted as indicated (**F**), with the dotted line in F indicating where the image was cropped to remove an irrelevant lane. Raw data for the full image are provided in the Source Data file. Relative CAMK2 activation, as determined by T287 phosphorylation, was then quantified (**G**). Data are expressed relative to vector control which was arbitrarily set at 1. All data in (**B**, **D** and **E**) are representative of at least $n = 2$ independent experiments. In (**G**), data are presented as mean values +/− SEM from $n = 3$ independent experiments. NS indicates $p > 0.05$. Data were analysed by one-way ANOVA with Tukey's multiple comparisons test. Source data are provided as a Source Data file.

## Discussion

The pseudokinase scaffold PEAK1 has emerged as a critical regulator of intracellular signalling, transducing and integrating pleiotropic signals downstream of a variety of growth factor receptors and integrins. For example, PEAK1 promotes Ras activation via Grb2 binding[6], and couples to paxillin regulation at focal adhesions via Crk[8]. In addition, it can indirectly impact Csk and Stat3 via association with PEAK2[16] and ASAP1, PYK2 and Cbl via PEAK3[4]. Our discovery that PEAK1 can also promote $Ca^{2+}$ signals via PLCγ1 and moreover, binds and sustains activation of the $Ca^{2+}$ effector CAMK2 greatly expands

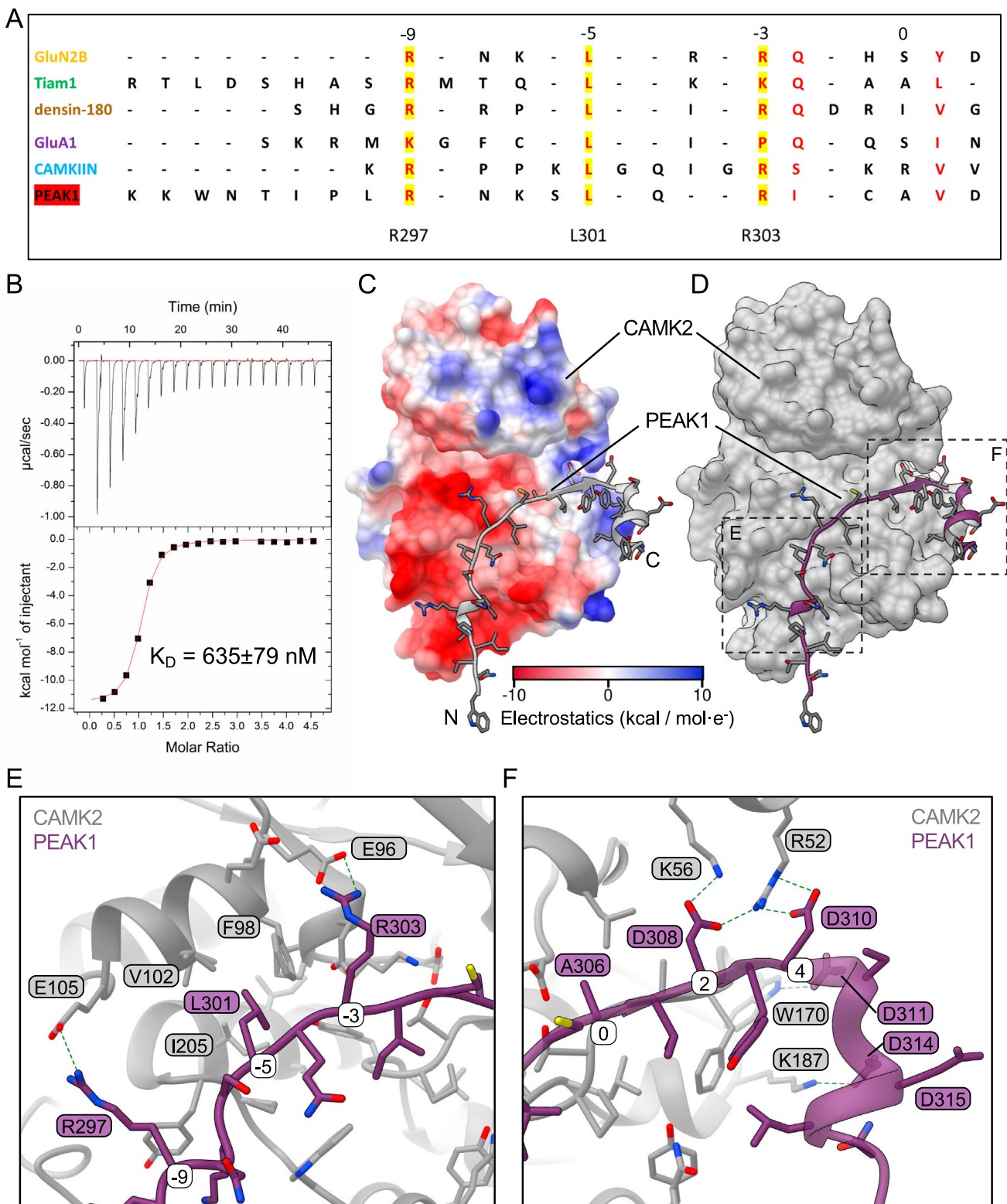

**Fig. 4 | Characterisation of a conserved CAMK2 interaction motif (CIM) in PEAK1. A** Sequence alignment of the PEAK1 CIM with other CAMK2 binding partners. The critical residues for CAMK2 interaction (R297, L301 and R303 in PEAK1) are highlighted in yellow. The canonical motif numbering (A306 being 0) is shown above the alignment. **B** Representative ITC measurement of CAMK2 kinase domain binding to PEAK1 CIM peptide. The mean $K_D$ value from $n = 3$ independent experiments is indicated. **C** Comparative modelling of the PEAK1/CAMK2 interaction by AlphaFold. Surface representation of CAMK2 with PEAK1 amino acids 291–320 shown as ribbon (grey). The orientation is as in Supplementary Fig. 6.

CAMK2 is coloured by electrostatic charge. The PEAK1 binding site on CAMK2 is noticeably negative, with key arginine residues of PEAK1 mediating interactions with CAMK2. The N- and C-termini are labelled as N and (**C**), respectively. **D** Surface representation of modelled PEAK1/CAMK2 interaction. CAMK2 is grey and PEAK1 amino acids 291–320 are purple. **E, F** Focused view of key predicted molecular interactions. Boxed regions from (**D**) illustrating predicted formation of salt bridges and electrostatic interactions (dotted lines) by specific PEAK1 residues across the surface of CAMK2. The canonical numbering of the PEAK1 motif is shown in white boxes. Source data are provided as a Source Data file.

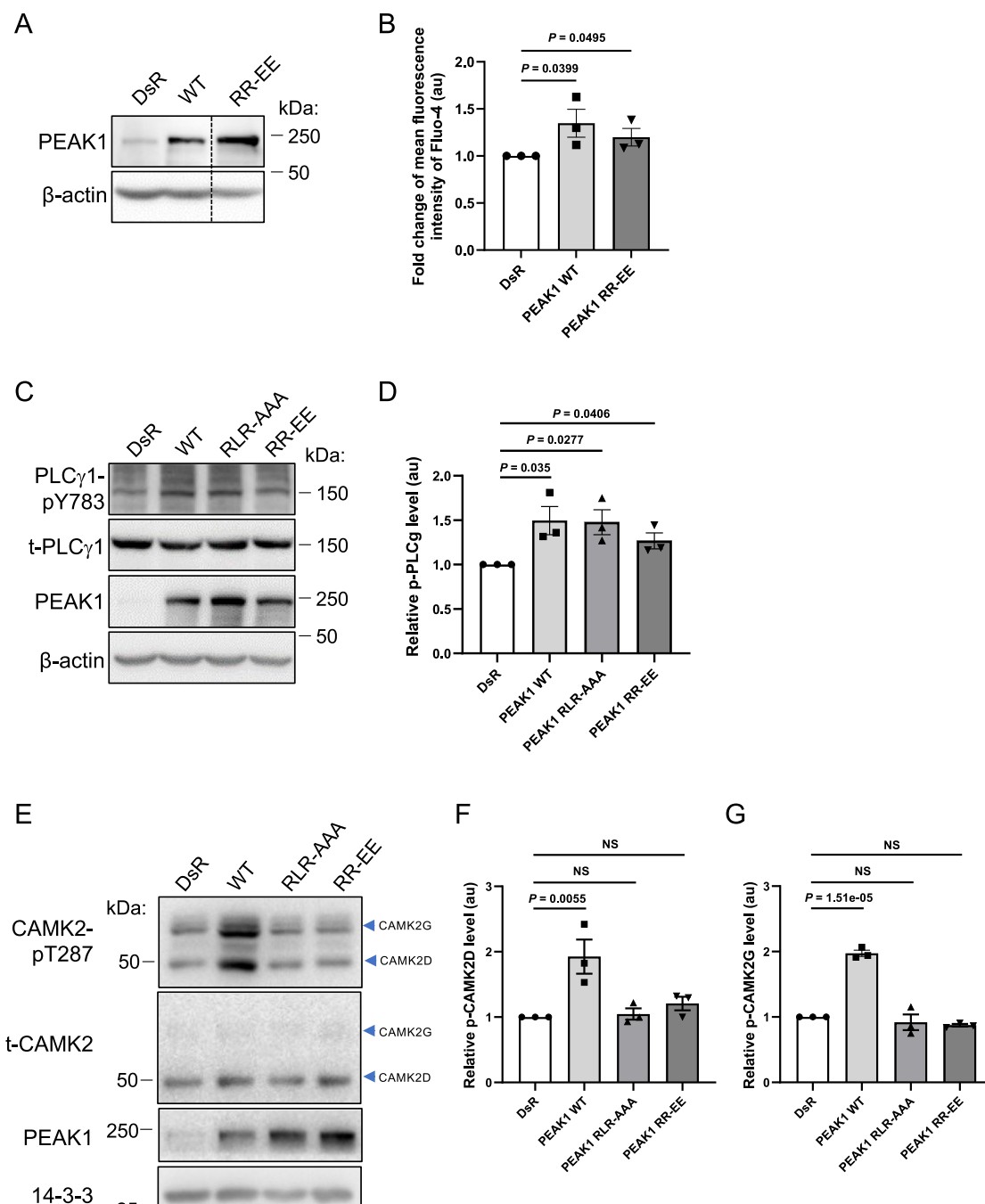

**Fig. 5 | The role of the PEAK1 CIM in CAMK2 activation. A**, **B**. Effect of CIM mutation on intracellular $Ca^{2+}$ levels. Constructs expressing WT PEAK1 or the PEAK1 RR-EE mutant were stably expressed in MDA-MB-231_EcoR cells via retroviral transduction. PEAK1 expression levels were determined by Western blot (**A**) and $Ca^{2+}$ levels (**B**) assayed as in Fig. 2G-H. In (**A**), the dotted line indicates where the image was cropped to remove an irrelevant lane. Raw data for the full image are provided in the Source Data file. Data in (**B**) represent mean Fluo-4 fluorescence ± SEM from $n = 3$ independent experiments. Statistical analysis utilised an unpaired one-tailed $t$ test. **C**, **D**. Effect of CIM mutation on tyrosine phosphorylation of PLCγ1. The indicated plasmids were transiently transfected into HEK293 cells and cell lysates Western blotted as indicated (**C**). PLCγ1 tyrosine phosphorylation was normalised for total expression (**D**). Data in (**D**) are expressed relative to the vector control, which was arbitrarily set at 1 and represents the mean ± SEM of $n = 3$ independent experiments. Data were analysed by unpaired two-tailed $t$ test. **E**–**G** Effect of CIM mutation on CAMK2 activation. Plasmids encoding WT PEAK1 and the indicated mutants were transfected into MDA-MB-231 cells. Total cell lysates were Western blotted as indicated (**E**) and then relative CAMK2 activation was quantified (**F**, **G**). Data from $n = 3$ independent experiments are expressed relative to the vector (DsR) control which was arbitrarily set at 1. Data represent the mean ± SEM of $n = 3$ independent experiments. NS indicates $p > 0.05$. Data were analysed by one-way ANOVA with Dunnett's multiple comparisons test. Source data are provided as a Source Data file.

the signalling repertoire of this scaffold and cements its role as a fundamental cell signalling nexus. It also reveals PEAK1 as a kinase superfamily member that sustains activation of CAMK2 via direct binding and an oncoprotein that targets CAMK2 via this mechanism. This expansion of the known signalling functions of PEAK1 calls for a re-evaluation of its role in physiological processes linked to CAMK2 signalling including learning and memory, cardiac function and osteogenic differentiation[20]. As discussed later, it also highlights a potential strategy for therapeutically targeting the oncogenic activity of PEAK1.

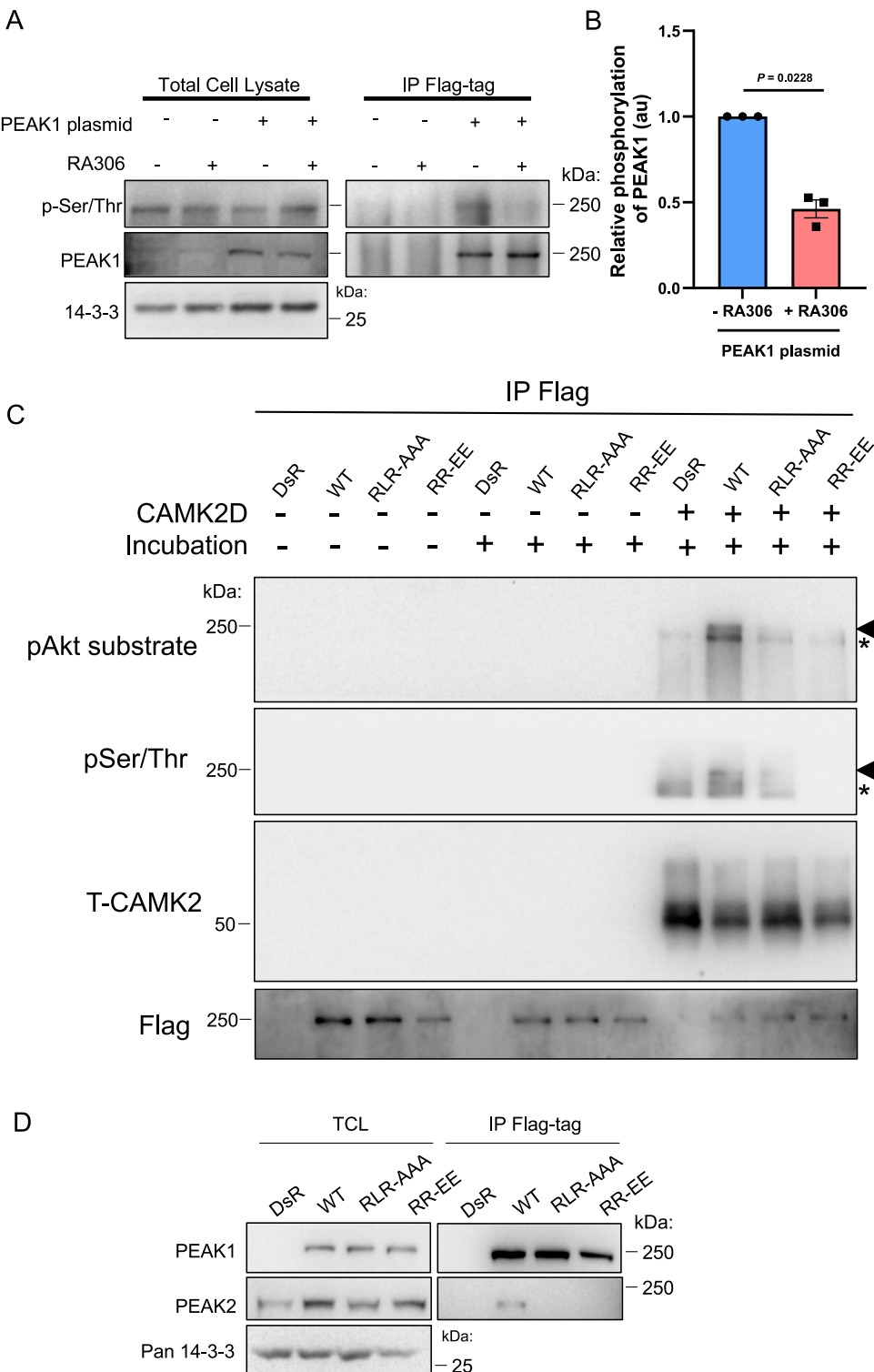

**Fig. 6 | CAMK2 phosphorylates PEAK1 to regulate the PEAK1 interactome. A, B.** RA306 inhibits PEAK1 serine/threonine phosphorylation. MDA-MB-231_EcoR cells stably expressing Flag-tagged PEAK1 were treated with RA306 (1 μM) for 24 h and cell lysates were directly Western blotted or subject to immunoprecipitation with Flag antibody prior to blotting as indicated (**A**). Relative serine/threonine phosphorylation of PEAK1, normalised for total PEAK1, is shown in the histogram, which presents the mean ± SEM of *n* = 3 independent experiments (**B**). In panel (**B**), data were analysed by ratio paired two-tailed *t* test. **C** CAMK2 phosphorylates PEAK1 in vitro. Flag-tagged versions of WT and mutant PEAK1 proteins were expressed in HEK293T cells. PEAK1 proteins were immunoprecipitated using anti-Flag antibody and following washing, immediately treated with SDS-PAGE sample buffer (SB) (no incubation) or incubated in kinase reaction buffer supplemented as indicated at 37 °C for 30 min. Reactions were terminated with SB. Samples were then Western blotted as indicated. The asterisk indicates a background band present in all CAMK2D-incubated samples, including the vector control while the arrowhead highlights phosphorylated PEAK1. **D** Impact of CIM mutations on heterotypic association with PEAK2. Flag-tagged WT PEAK1 or PEAK1 mutants were expressed in MDA-MB-231 cells and association with endogenous PEAK2 assayed by IP/Western. Data in (**C**, **D**) are representative of *n* = 3 independent experiments. Source data are provided as a Source Data file.

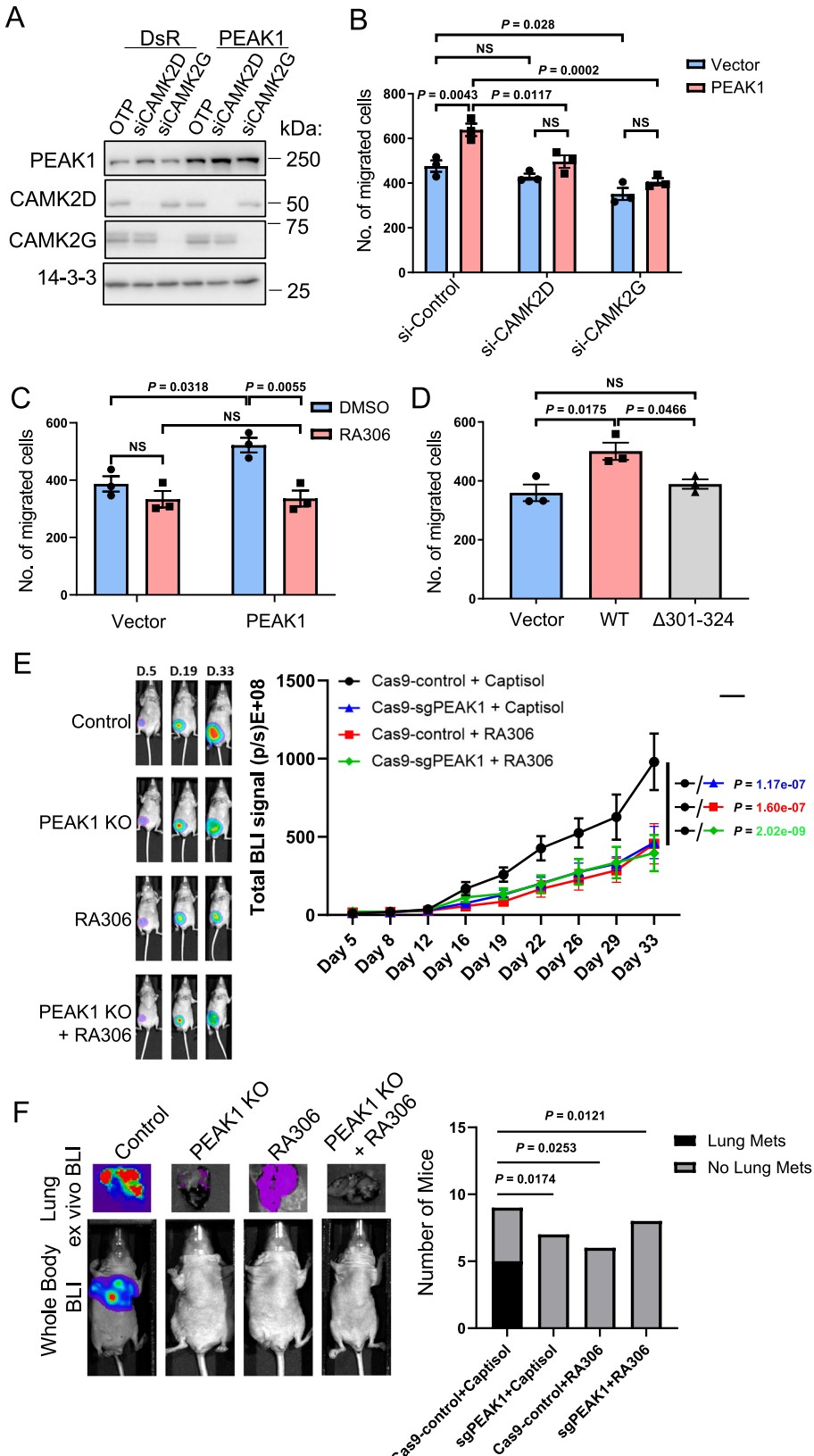

Our findings lead to a model whereby PEAK1 enhances and prolongs CAMK2 activation and signalling by two mechanisms that together provide a 'feed-forward' regulatory process for CAMK2 (Fig. 8). First, PEAK1 overexpression leads to enhanced and more sustained $Ca^{2+}$ signals in cells, reflecting increased tyrosine phosphorylation and hence activation of PLCγ1 which is driven in an SFK-dependent manner. While the role of SFKs in this pathway is consistent with PEAK1 functioning within an SFK signalling network in TNBC[27], the detailed mechanism currently remains unclear and based on our BiCAP data, does not appear to involve direct binding of PEAK1/2 to SFKs or PLCγ1. In this regard, it differs from the direct binding and regulation of the tyrosine kinases Csk and PYK2 by PEAK2 and PEAK3, respectively[12,34]. In terms of

**Fig. 7 | Role of CAMK2 in PEAK1-regulated TNBC biology. A** PEAK1 over-expression with CAMK2D and CAMK2G knockdown in MDA-MB-231 cells. Cells were transfected with a PEAK1 plasmid +/− siRNA-mediated CAMK2D or CAMK2G knockdown. Cell lysates were Western blotted as indicated. Data are representative of $n = 3$ independent experiments. **B** Role of CAMK2D/G in PEAK1-promoted MDA-MB-231 cell migration. Cells validated in (**A**) were subjected to a transwell migration assay. **C**, **D** Role of CAMK2 activation. MDA-MB-231 cells were transfected with a PEAK1 plasmid and subject to transwell migration assays +/− RA306 (**C**). Plasmids expressing WT PEAK1 and the Δ301–324 mutant that cannot activate CAMK2 were transiently transfected into MDA-MB-231 cells, and cells subjected to a transwell migration assay (**D**). **E** Role of PEAK1/CAMK2 signalling in TNBC xenograft growth. Control MDA-MB-231_HM cells (Cas9-control) or PEAK1 KO cells (Cas9-sgPEAK1) in combination with RA306 or vehicle control (captisol) were injected into the mammary fat pad. Left panel, representative whole body bioluminescent imaging (BLI) images. Right panel, quantification of xenograft growth. Each treatment

group exhibited 9 mice. **F** Role of PEAK1/CAMK2 signalling in TNBC metastasis. Control MDA-MB-231_HM cells (Cas9-control) or PEAK1 KO cells (Cas9-sgPEAK1) in combination with RA306 or vehicle control (captisol) were injected into the mammary fat pad. At Day 19 post-injection with primary tumour size at ~100 mm³, they were resected, and tumour growth at secondary sites measured. Left panel, representative whole body and ex vivo lung BLI images from each group at Day 72 post-injection. Right panel, comparison of obvious metastatic growth (defined as BLI > 500 E + 05 p/s) across the different treatment groups. Data are presented as mean values +/− SEM from $n = 3$ independent experiments (**B**–**D**) or from the individuals of each group (**E**). In (**F**), mouse numbers were: Cas9 control + captisol, 9; sgPEAK1 + captisol, 7; Cas9 control + RA306, 6; sgPEAK1 + RA306, 8. NS indicates $p > 0.05$. Data were analysed by two-way ANOVA with Tukey's (**B**, **C**) or Dunnett's (**E**) multiple comparisons tests, one-way ANOVA with Tukey's multiple comparisons test (D), or Chi-square test (**F**). Source data are provided as a Source Data file.

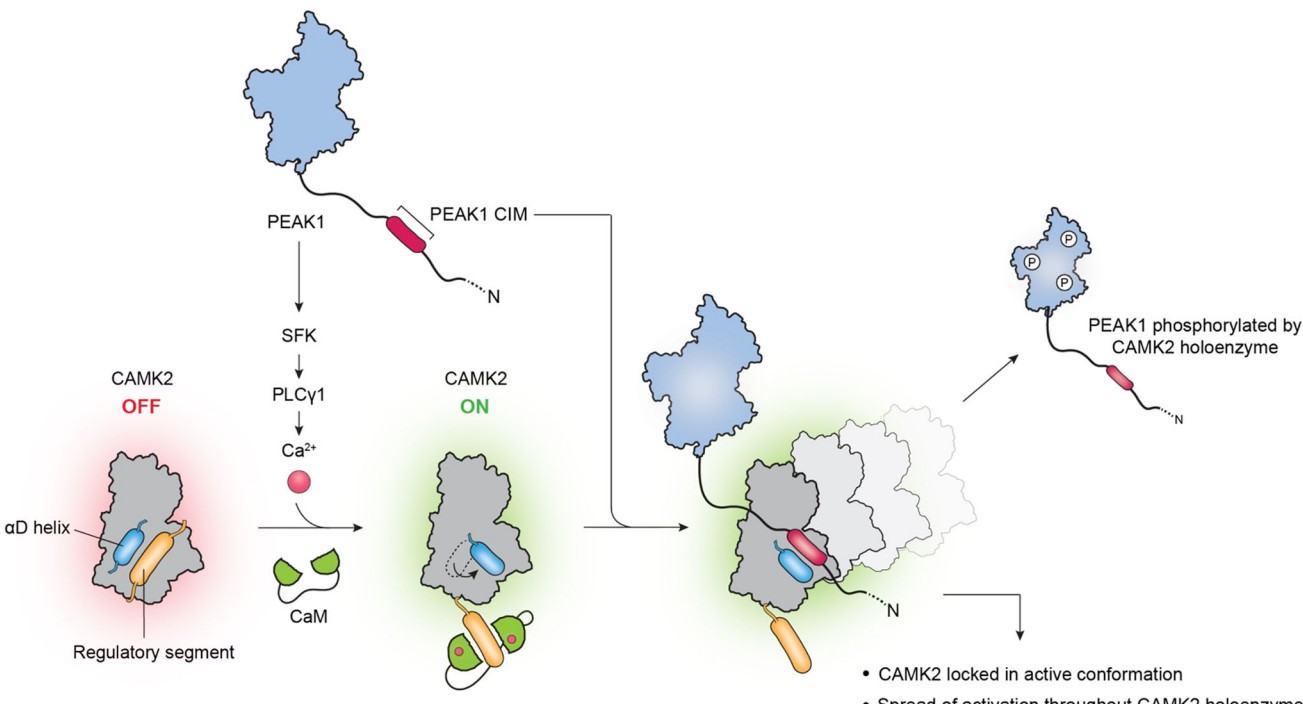

**Fig. 8 | Schematic representation of feed-forward regulation of CAMK2 by PEAK1.** PEAK1 enhances intracellular Ca²⁺ by promoting SFK-mediated tyrosine phosphorylation of PLCγ1. This leads to classical activation of CAMK2 by Ca²⁺/CaM and the αD helix rotates out. The PEAK1 CIM then binds to the CAMK2 substrate binding site and competes with the regulatory segment, effectively 'locking' CAMK2 in an active conformation. Sustained phosphorylation of PEAK1 and CAMK2

itself (on T287) then occurs as the PEAK1 CIM undergoes cycles of dissociation and re-association. However, other mechanisms may contribute to the observed effects on CAMK2 activation, as indicated on the figure. CIM-mediated recruitment of PEAK1 to the CAMK2 holoenzyme promotes serine/threonine phosphorylation of PEAK1 which in turn positively regulates heterotypic association with PEAK2.

the SFKs responsible, TNBCs express several SFKs including Src, Yes, Fyn and Lyn, but expression of the latter is particularly high, and since Lyn also regulates PEAK1 tyrosine phosphorylation, it represents an interesting candidate for further investigation[6,27,35]. The enhanced Ca²⁺ signal generated by this PEAK1/SFK/PLCγ1/Ca²⁺ pathway leads to a greater pool of activated CAMK2. Second, PEAK1 binds directly to the Ca²⁺-activated CAMK2 via the conserved CIM and this has two effects – first, it leads to sustained CAMK2 activation, as determined by T287 phosphorylation, and second, it promotes CAMK2-mediated PEAK1 phosphorylation. We note that sustained elevation of intracellular Ca²⁺ can be pro-apoptotic[20], but this is likely countered by survival signals emanating from PEAK1, for example via Grb2/Ras/Erk[5].

Promotion of CAMK2 T287 phosphorylation and PEAK1 phosphorylation by binding of the PEAK1 CIM to CAMK2 can be explained

by a model that incorporates two key characteristics of this enzyme/interactor system - first, the known ability of CAMK2 to form a large holoenzyme comprising 12–14 monomers[24] and second, PEAK1 CIM binding 'locking' CAMK2 in an active state with the αD helix rotated outward[30] (Fig. 8). In this model, repeated cycles of association/dissociation of one PEAK1 CIM to different CAMK2 subunits within the complex leads to sustained activation of a subset of subunits, allowing continued autophosphorylation on T287, which occurs in trans [24]and phosphorylation of PEAK1. However, we accept that alternative, or contributory, mechanisms are conformational changes within the holoenzyme induced by PEAK1 binding to one monomer that promotes more widespread enzyme activation and autophosphorylation, and T287 being protected from phosphatase action by PEAK1 binding to CAMK2. In addition, CIM-mediated recruitment of PEAK1 to one

monomer within the $Ca^{2+}$-activated oligomer leads to phosphorylation of PEAK1 by other activated monomers within the complex. This would be facilitated by the relatively large number of consensus CAMK2 phosphorylation sites present on PEAK1 (41 R/KXXS/T sites in total). Importantly, a precedent for a reciprocally-activating interaction of CAMK2 with a binding partner exhibiting a pseudosubstrate CIM is provided by CAMK2/TIAM1. Here, binding of the TIAM1 CIM to CAMK2 leads to sustained phosphorylation of TIAM1 by CAMK2[31].

An interesting question is the stoichiometry of the resulting PEAK1/CAMK2 complexes. While CAMK2 forms a large holoenzyme complex of 12–14 subunits, PEAK1 can form dimers via the SHED domain and also higher-order oligomers via pseudokinase domain interfaces[18]. These properties may facilitate the formation of assemblies with CAMK2 oligomers. In addition, the localisation of the PEAK1 CIM towards the N-terminus of a largely disordered extension provides considerable flexibility for the assembly of large multimeric protein complexes. Interestingly, in contrast to PEAK1, PEAK2 lacks a consensus CIM, and its dimerisation potential is essential for CAMK2 association. While the impact of PEAK2 on CAMK2 regulation requires further characterisation, one possibility is that PEAK2 contributes to the assembly of higher-order complexes of PEAK1/2 with CAMK2.

While PEAK1 is known to be regulated by site-selective tyrosine phosphorylation[6,19,36], our study demonstrates positive regulation of PEAK1 by serine/threonine phosphorylation. Furthermore, we determine that like tyrosine phosphorylation, this regulates the PEAK1 interactome. In particular, the two PEAK1 CIM mutants, which are poor substrates for CAMK2, exhibit decreased association with PEAK2. This provides one explanation for the role of CAMK2 in PEAK1-enhanced migration since heterotypic interaction with PEAK2 is required for this process[16]. Conceivably, CAMK2-mediated phosphorylation may regulate PEAK1 conformation and accessibility of the dimerisation and/or oligomerization interfaces. Another protein that exhibits significantly reduced binding to the PEAK1 CIM mutants is 14-3-3. This is consistent with the overlap between the minimal CAMK2 phosphorylation site consensus (R/KXXS/T) and the 14-3-3 mode 1 motif for binding, RXXpS/TXP. In the case of PEAK3, 14-3-3 binding to phosphorylated PEAK3 S69 impairs Grb2 and Crk recruitment to the scaffold, and the PEAK3 S69A mutant exhibits a markedly enhanced ability to promote cell migration[23]. With PEAK1, the overall effect of CAMK2-mediated phosphorylation on PEAK1-promoted migration/invasion is positive, so 14-3-3 binding may downregulate binding of negative regulatory factors, and/or increase association with positive ones, such as PEAK2.

Reflecting their role in a variety of disease states, pseudokinases have emerged as potential therapeutic targets, albeit challenging ones given their lack of catalytic activity. Possible therapeutic intervention strategies include the use of small molecule allosteric regulators, therapeutic antibodies for cell surface receptor pseudokinases such as HER3, targeted degradation approaches e.g., using PROTACs, and pharmacologic modulation of 'actionable' downstream effector pathways[3]. The oncogenic pseudokinase PEAK1 presents a particular conundrum since it does not bind ATP or GTP, the dimerisation interface is relatively large and difficult to target using small molecule drugs, and PEAK1 is an intracellular protein and, therefore, not accessible by standard therapeutic antibody approaches[5]. However, in this study, we identify a strategy for targeting PEAK1 signalling that exploits the functional dependency of this pseudokinase on a readily 'actionable' bona fide kinase, CAMK2. Importantly, supporting therapeutic targeting of this serine/threonine kinase, CAMK2 isoforms are overexpressed in a variety of human cancers, including breast cancer, where they promote cell proliferation, migration and invasion[20,21]. They are also positively associated with metastasis. In an animal model of prostate cancer metastasis, genetic ablation of all CAMK2 family members significantly reduced lymph node metastasis[37]. Expression, as well as activation, of CAMK2 is significantly higher in lymph node metastases than in primary breast cancers, and high expression of the

CAMK2 family is associated with worse DMFS in TNBC[21]. In addition, brain metastatic colonisation of TNBC is associated with CAMK2 signalling[38]. Our work now provides important insights into the oncogenic mechanism of CAMK2, with a key role being CAMK2-mediated phosphorylation of PEAK1. This promotes heterodimerization of PEAK1 with PEAK2, a process required for PEAK1 to promote cell migration, presumably because PEAK2 recruits/activates critical effectors such as Csk and Stat3[16]. In addition, the activated CAMK2 may phosphorylate other downstream targets that promote metastasis, such as the Rho family GEF TIAM1[31], and regulate the focal adhesion proteins FAK and paxillin[39].

In general, characterisation of the oncogenic role of CAMK2 has been compromised by a poor understanding of regulatory mechanisms, with functional studies often utilising constitutively active mutants, as well as a reliance on tool compounds for CAMK2 inhibition, such as KN93, that have numerous off-target effects[40]. Our study addresses these issues by identifying the oncogenic pseudokinase PEAK1 as a feed-forward activator of CAMK2 and the second-generation CAMK2D/G inhibitor RA306 as a suitable targeted agent to block PEAK1/CAMK2 signalling in TNBC and other cancers where PEAK1 is involved, such as pancreatic cancer. Importantly, RA306 avoids central nervous system side effects and exhibits good oral bioavailability, highlighting its potential for oncology re-purposing[32]. In addition, the identification of a reciprocally activating PEAK1/CAMK2 complex indicates that these two proteins could be used as predictive biomarkers to guide patient stratification for anti-CAMK2 treatment. Overall, our work identifies an unsuspected interaction across the kinase superfamily with significant implications for our understanding of cell signalling in normal and disease states and precision targeting of PEAK1 and CAMK2.

## Methods

### Research ethics
All research complied with ethical guidelines established by the National Health and Medical Research Council (NHMRC). Ethics approval for animal work is described in the corresponding section.

### Plasmids and siRNAs
The retroviral expression vector pRetroX-IRES-DsRedExpress (DsR) containing N-terminal HA-tagged and C-terminal FLAG/Myc-tagged PEAK1 as well as its deletion mutants including ΔNterm, ΔCterm and ΔN1, together with the expression vector pMIG containing N-terminal HA-tagged and C-terminal FLAG-tagged PEAK2 as well as PEAK2 site mutants (L955A, L966A, W1382A) were previously generated in our laboratory[16,18]. pRetroX-IRES- DsRedExpress plasmid containing Flag-tagged PEAK1 N terminal (residues 1–324) deleted and other mutants (Δ1–260, Δ261–300, Δ261–324, Δ301–324, R297A/L301A/R303A and R297E/R303E) were purchased from GenScript.

pDEST-V1 (Addgene plasmid 73637) and pDEST-V2 (Addgene plasmid 73638) vectors were assembled from various recombinant and synthetic components as described previously[22]. Vectors expressing V1- or V2-labelled fusions of PEAK1 or PEAK2 were generated by recombination cloning from pDONR223-PEAK1 and pDONR223-PEAK2 constructs into pDEST-V1 or pDEST-V2 destination vectors using Gateway LR Clonase enzyme mix (Life Technologies) according to the manufacturer's instructions and verified by sequencing. The expression vector encoding full-length Venus fluorescent protein was a gift from S. Michnick (University of Montreal).

CAMK2A kinase domain (residues 7–274) was cloned into a pET vector containing an N-terminal 6 × His followed by a SUMO tag. A D135N mutation was introduced by site-directed mutagenesis.

Sequences of siRNAs were PEAK1 (ON-TARGETplus Human PEAK1 siRNA, cat. LQ-005339-00-0010), PEAK2 (MISSION Human PEAK2 siRNAs, cat. #1: SASI_Hs02_00316033, #2: SASI_Hs02_00316034, #3: SASI_Hs02_00316035, #4: SASI_Hs02_00316036), CAMK2D

(siGENOME Human CAMK2D siRNAs, cat. #1: D-004042-02-0002, #2: D-004042-03-0002, #3: D-004042-04-0002, #4: D-004042-05-0002), CAMK2G (siGENOME Human CAMK2G siRNAs, cat. #1: D-004536-01-0005, #2: D-004536-03-0005, #3: D-004536-05-0005, #4: D-004536-06-0005). Negative control siRNA was ON-TARGET plus Non-targeting pool (Dharmacon, D-001810-10-20).

## Antibodies

The following antibodies were purchased from Santa Cruz Biotechnology (Dallas, TX): PEAK1 (cat. sc-100403), β-actin (cat. sc-69879) and Pan 14-3-3 (cat. sc-1657). CAMK2D (cat. ab181052) and CAMK2G (cat. ab201966) were purchased from Abcam. Phospho-Akt Substrate (RXXS*/T*) (cat. 9614), T-CAMK2 (cat. 4436), CAMK2-pT287 (cat. 12716), PLC-γ (cat. 5690S), PLC-γ-pY783 (cat.14008S) and cleaved PARP Asp 214 (cat. 9546) were purchased from Cell Signalling Technology. Flag (cat. F1804 and F7425), and α-tubulin (cat. T5168) were purchased from Sigma. The following antibodies were also used: CAMK2D (Genetex, cat. GTX111401), HA (Roche Applied Science, cat. 11867423001). Phospho-Ser/Thr antibody (cat. 612548) was purchased from BD Transduction Laboratories. The custom-made PEAK2 (SgK223) antibody was generated by Cambridge Research Biochemicals Ltd. by standard procedures and described previously[14].

## Inhibitors and chemicals

RA306 was custom synthesised by Reagency (cat. RGNCY-0117). The SFK inhibitor eCF506 was obtained from Professor Margaret Frame (Edinburgh Cancer Research Centre). Captisol (cat. A508148) was purchased from AmBeed. BAPTA-AM (cat. A1076) and DMSO (cat. d8418) were purchased from Sigma. Doxorubicin-HCl was obtained from Selleckchem (cat. S1208).

## Cell lines and tissue culture

HEK293T cells were maintained in Dulbecco's Modified Eagle's Medium (DMEM, Gibco, cat. 1200046) supplemented with 10% (v/v) foetal bovine serum (FBS; Fisher Biotech, cat. S-FBS-US-015). MDA-MB-231 parental cells were obtained from EG&G Mason Research Institute, Worcester, MA. MDA-MB-231 cells stably expressing the murine ecotropic receptor (MDA-MB-231_EcoR) were a kind gift from A/Prof Maija Kohonen-Corish at the Garvan Institute of Medical Research. MDA-MB-468 cells were obtained from the American Type Culture Collection (ATCC, Number HTB-132). The highly metastatic MDA-MB-231 cell line (MDA-MB-231_HM) was originally established by Prof. Zhou Ou from Fudan University Shanghai Cancer Centre in China. The MDA-MB-231_HM cell line expressing luciferase/mCherry was a kind gift from A/Prof. Erica Sloan at Monash University. In this paper, MDA-MB-231_HM refers to the cell line expressing luciferase/mCherry.

Breast cancer cell lines were maintained in RPMI-1640 medium (Life Technologies, cat. 11875119 and Gibco, cat. 31800-089) supplemented with 10% FBS (Fisher Biotec, cat. S-FBS-US-015), 10 μg/mL Actrapid penfill insulin (Clifford Hallam Healthcare), and 20 mM HEPES (Life Technologies, cat. 15630080). All cell lines were authenticated by short tandem repeat polymorphism, single-nucleotide polymorphism, and fingerprint analyses and underwent routine mycoplasma testing by PCR.

## Cell lysis, immunoprecipitation, and immunoblotting

Cell lysates for immunoblotting and immunoprecipitation (IP) were prepared using radioimmunoprecipitation (RIPA) buffer and normal lysis buffer (NLB), respectively[41]. For IP of overexpressed Flag- or HA-tagged proteins, cell lysates were incubated with either anti-FLAG M2 affinity-agarose beads (Sigma, cat. A2220) or red anti-HA affinity gel (Sigma, cat. E6779) at 4 °C for 2–4 h with end-to-end rotation. After the incubation, the beads were washed extensively with ice-cold lysis buffer, the immune complexes were eluted with SDS-PAGE loading buffer, and subjected to Western blotting analysis, which was performed following SDS-polyacrylamide gel electrophoresis using 5% stacking gels and 8%, 10% or 12% separating gels. In order to ensure that phospho and total antibodies were used at non-limiting concentrations, Western blot analyses were undertaken on a dilution series of cell lysates. Uncropped and unprocessed scans of blots are provided in the Source Data file or in the Supplementary Information.

## Generation of PEAK1 knock-out cells

Stable MDA-MB-231_HM PEAK1 knockout cells were generated using the Synthego Immortalised Cell Nucleofection Protocol as per the manufacturer's instructions (Synthego). Briefly, ribonucleoprotein (RNP) complexes were assembled by combining Cas9 2NLS (Synthego), human PEAK1 sgRNA (Gene Knockout Kit v2, Synthego) and Nucleofector SolutionTM L + supplement (Synthego) and incubating at room temperature for 10 mins. RNP complexes were combined with cells in NucleocuvetteTM strips immediately prior to electroporation. Cells were electroporated using the 4D-X Core unit (Lonza) using setting CH-125. Cells were immediately resuspended in a normal culture medium following electroporation. Cell lysates were collected 72 h post-electroporation to confirm PEAK1 knockout efficiency by Western blot.

## Transfections

SMARTpool siRNA or individual siRNA duplexes were applied to cells using DharmaFECT (Dharmacon, Lafayette, CO) transfection reagent according to the manufacturer's instructions. Plasmid transfections were performed using Lipofectamine 3000 (Invitrogen) according to the manufacturer's instructions.

## BiCAP purification

This was performed as previously described[22]. Briefly, HEK293T cells were transfected with the appropriate BiCAP constructs. Cells were lysed with ice-cold lysis buffer comprising 50 mM Tris (pH 7.4), 150 mM NaCl, 1 mM EDTA, 1% (v/v) TritonX-100 supplemented with fresh EDTA-free protease inhibitor cocktail and 0.2 mM sodium orthovanadate. Samples were cleared by centrifugation at $18,000 \times g$ for 10 min at 4 °C to remove cellular debris. Total protein concentration was determined by Bradford assay using Bio-Rad Protein Assay Dye Reagent Concentrate (Bio-Rad Laboratories) according to the manufacturer's instructions. To purify BiCAP dimers (or Venus alone as a negative control for nonspecific interactions), 1 mg of total lysate was incubated with 35 μl of GFP-Trap_A agarose beads slurry (ChromoTek GmbH) for 2 h at 4 °C with end-to-end rotation. Beads were extensively washed with wash buffer [10 mM Tris (pH 7.5), 150 mM NaCl, 0.5 mM EDTA] and subjected to on-bead digestion with trypsin, and the resulting peptides were analysed by mass spectrometry (MS). Samples from each experimental condition were subjected to data-dependent acquisition (DDA) to construct a spectral library. Independent experiments ($n = 3$) for each condition were then analysed by data-independent acquisition (DIA) to characterise the corresponding interactomes. Conditions for DDA and DIA are detailed below.

## Mass spectrometry-based proteome analysis

MS analysis was based on an UltiMate 3000 RSLC nano-LC system (Thermo Fisher Scientific) coupled to a Q Exactive Plus Mass Spectrometer (Thermo Fisher Scientific). The peptides were loaded via a Pepmap100 trap column, 100 μm × 2 cm nanoviper, (Thermo Fisher Scientific), eluted and separated on a Pepmap100 RSLC analytical column, 75 μm × 50 cm, nanoViper, C18 (Thermo Fisher Scientific). For each injection, 1 μg of peptides was loaded onto the column and eluted using increasing concentrations of buffer B (80% acetonitrile / 0.1% formic acid) at 250 nL/min over 150 min. The eluent was nebulised and ionised using a nanoelectrospray source (ThermoFisher Scientific) with a distal-coated fused silica emitter (Trajan, Ringwood, Victoria, Australia). The capillary voltage was set at 1.7 kV.

To generate spectral libraries for proteomics, the MS was firstly operated in DDA mode to automatically switch between full MS scans and tandem MS/MS scans. Survey full scan MS spectra (m/z 375–1700) were acquired in the Orbitrap with 70 000 resolution after accumulation of ions to a $1 \times 10^6$ target value with a maximum injection time of 120 ms. Up to 12 of the most intensely charged ions ($z \geq +2$) were sequentially isolated and fragmented in the collision cell by higher-energy collisional dissociation with the following parameters, fixed injection time of 120 ms, 35 000 resolution and automatic gain control target of $5 \times 10^5$. Dynamic exclusion was set to 15 s.

The DIA mode consisted of an MS1 scan (scan range from 400 to 1220 m/z, at the resolution of 35 000, an AGC target of $5 \times 10^6$ and a maximum ion injection time of 120 ms) followed by MS2 scans with 60 DIA windows at a resolution of 35 000, an AGC target of $3 \times 10^6$ with automatic injection time, isolation window of 30 m/z. The stepped collision energies were 22.5%, 25%, and 27.5%. The spectra were recorded in profile type.

### Data processing
The DDA spectra were analysed with MaxQuant (version 1.5.2.8)[42] against the human UniProt fasta database (v2015–08, 20,210 entries) using default settings (protease = Trypsin/P; Maximum missed cleavages = 2; fixed modifications = Carbamidomethyl (C); variable modification = Acetyl (Protein N-term);Oxidation (M); minimum peptide length = 7; mass tolerance for precursor = 4.5ppm; minimum number of unique peptides for protein identification = 1; peptide- and protein-level false discovery rate (FDR) = 1%). The spectral library was then generated in Spectronaut® software (version 8, Biognosys, Schlieren, Switzerland) and normalised to iRT peptides. The DIA data were firstly transformed into htrms format and analysed with Spectronaut® using default settings (XIC RT Extraction Window = Dynamic; Calibration Mode = Automatic; iRT Calibration Strategy = Non-linear iRT calibration; Interference Correction = True; Profiling Strategy = iRT Profiling; Profiling Row Selection = Minimum Qvalue Row Selection; peptide- and protein-level FDR = 1%). The MS data were searched against the same human UniProt database and the aforementioned spectral library. The false discovery rate (FDR) was set to 1% at peptide and protein levels. Perseus (version 2019) was used to normalise the data and impute for missing values[43]. Candidate interacting proteins were defined by applying the following criteria: *p*-value < 0.05 by two-tailed *t* test and a log2 fold change of ≥ 1.5 against the Venus control. Pathway enrichment was undertaken using the WikiPathways Database.

### Protein expression and purification
CAMK2A D135N kinase domain was recombinantly expressed in BL21(DE3) cells (Millipore) as previously described[30]. Protein expression was induced for ~6 h by 1 mM isopropyl β-a-D-1-thiogalactopyranoside (IPTG; GoldBio) in an 18 °C shaker. Cell pellets were resuspended and lysed in Buffer A [25 mM Tris pH 8.5, 50 mM KCl, 40 mM imidazole, 10% glycerol] with protease inhibitor mixture [0.2 mM AEBSF, 5.0 μM leupeptin, 1 μg/ml pepstatin, 1 μg/ml aprotinin, 0.1 mg/ml trypsin inhibitor, 0.5 mM benzamidine] and DNase (1 μg/ml; Sigma). Protein purification was performed using an ÄKTA pure chromatography system at 4 °C. After pelleting the cell debris, clarified cell lysate was loaded onto two 5 mL HisTrap FF Ni-NTA Sepharose columns (GE) and eluted with a combination of 50% buffer A and 50% buffer B [25 mM Tris-HCl pH 8.5, 150 mM KCl, 1 M imidazole, 10% glycerol] for a final concentration of 0.5 M imidazole. Protein was desalted using a HiPrep 26/10 desalting column (GE), and His-SUMO tags were cleaved with Ulp1 protease overnight at 4 °C in buffer C [25 mM Tris-HCl pH 8.5, 40 mM KCl, 40 mM imidazole, 2 mM tris(2-carboxyethyl) phosphine (TCEP), 10% glycerol]. Cleaved tags were separated by a subtractive Ni-NTA step followed by an anion exchange step performed with a 5 ml HiTrap Q-FF (GE) and protein was eluted with a KCl gradient. Eluted proteins were concentrated and further purified in gel

filtration buffer [25 mM tris-HCl pH 8.0, 150 mM KCl, 1 mM TCEP, 10% glycerol] using a Superdex 75 10/300 GL size exclusion column (GE). Pure fractions were then concentrated, aliquoted, flash-frozen in liquid nitrogen, and stored at − 80 °C until further use.

### Peptide synthesis
The PEAK1 peptide used for ITC was synthesised to > 95% purity with acetylation at the N-terminus and amidation at the C-terminus (Genscript). The peptide sequence was as follows:KKWNTIPLRNKSLQRICAVDYDDSYD.

### Isothermal titration calorimetry
ITC data were obtained using a Malvern Auto-iTC200 calorimeter (UMass Amherst). The peptide was dissolved in gel filtration buffer with 6% v/v DMSO and concentration was determined using ε280 = 8370 M-1cm-1. CAMK2 kinase domain was diluted in gel filtration buffer with 5.6% v/v DMSO and concentration was determined with ε280 = 43850 M-1cm-1. Titrations were performed with the peptide as the titrant. Blank titrations were performed by injecting the peptide into gel filtration buffer. All titrations were performed at 20 °C using the standard 19-injection method where one injection of 0.4 μL is followed by 18 × 2 μL injections with 150 s between injections. The stir speed was set to 750 rpm, and the reference power was set to 10. Data were buffer-subtracted and analysed using MicroCal ITC Origin add-on dissociation model v1.00 using the one-site fitting model.

### Comparative modelling of PEAK1(291-320):CAMK2
Several candidate structural models of PEAK1(291-320):CAMK2 were generated by AlphaFold-Multimer[44,45] (v3) using the ColabFold[46] (v1.5.1) community implementation. Multiple sequence alignments and template searches were performed by MMseqs[44] and HHsearch using default settings. Custom templates of CAMK2 in complex with known tight binders were required to achieve models with acceptable metrics of pLTTD and pTM scores. These interactions were manually curated from crystal structures determined by Özden et al.[30] (crystal symmetry mates were discarded). Six recycle rounds were conducted and final models were ranked according to pTMscore. The best scoring models were relaxed with Amber[47] to minimise steric clashes.

### Cross species multiple sequence alignment of PEAK1 N-terminus
To generate a multiple sequence alignment, two rounds of PSI-BLAST[48] were performed against the non-redundant clustered database[49] (to increase taxonomic depth) using human PEAK1 as a query sequence. A cutoff of the expectation value of $1 \times 10^{-3}$ was used to select sequences for subsequent rounds and alignment. Finally, the top 50 and top 250 sequences (sorted by expectation value) were subjected to alignment in Clustal Omega[50,51]. The full sequence range for the top 250 sequences was subjected to a MEME[52] search to identify consensus motifs.

### Proximity ligation assay (PLA)
PLA was performed to detect protein-protein interactions in MDA-MB-231 cells using Duolink™ In Situ Red Starter Kit (Sigma, cat. DUO92101). All the reagents, including probes, detection reagents and wash buffers were provided in this kit. On day 1, $6 \times 10^4$ cells were seeded per well in a 48-well plate and incubated at 37 °C overnight. On day 2, cells were gently washed 3 times with 1x phosphate-buffered saline [PBS; NaCl (136 mM), KCl (2.6 mM), Na2HPO4 (10 mM), KH2PO4 (1.76 mM)], and then fixed in 4% (v/v) paraformaldehyde (VWR, cat. ALFA43368.9 M) for 30 min on ice. Following that, cells were washed with 1xPBS again and permeabilized with 0.2% Triton X-100 (Sigma, cat. T8787) for 3 min at room temperature. After another 1xPBS washing step, cells were incubated with a blocking solution for 30 min at 37 °C, following which, cells were incubated with two primary antibodies from different species against the targeted proteins overnight at 4 °C. On day 3, cells were washed with 1x Tris-buffered saline (TBS)

solution with the detergent Tween-20 [TBS-T; Tris base (50 mM), NaCl (150 mM), pH 7.6, Tween-20 (0.01% v/v)], then PLA probes were added to the cells and incubated for 60 min at 37 °C. Following ligation and amplification steps, samples were mounted with Mounting Medium with DAPI (Invitrogen, cat. P-36931). Cells were photographed using a confocal microscope and dot numbers were analysed using ImageJ software (Version 1.52).

## Calcium assay

MDA-MB-231 EcoR cells expressing DsR empty vector, wildtype PEAK1 or PEAK1-CIM mutants were generated by retroviral-mediated transduction as previously described[53]. Cells were seeded into FluoroDish cell culture dishes (World Precision Instruments, cat. FD35-100) and cultured overnight. The medium was then replaced with starvation medium (RPMI-1640 medium supplemented with 0.4% FBS, 10 μg/mL Actrapid penfill insulin, 20 mM HEPES) supplemented with 5 μM Calcium indicator Fluo-4 AM (Invitrogen, cat. F14217). Cells were then incubated for 60 min at 37 °C. The dye-loading media was removed and replaced with a starvation medium without Fluo-4 AM. Images were acquired with a fluorescence microscope (Leica DMi8) using the FITC filter set with excitation peaking at 495 nm and emission peaking at 519 nm. The cellular fluorescence was quantitated using ImageJ software. A total of > 200 cells were quantified per condition in each biological replicate. The background was subtracted from the cell signal to plot fluorescence intensity.

## Proliferation assays

These were performed using the CellTiter-Glo® 2.0 Cell Viability Assay (Promega – G9241) as per the manufacturer's instructions. Briefly, MDA-MB-231 HM cells were harvested and seeded into High Attachment (HA) (Falcon, FAL353072) and ultra-low attachment (LA) (Corning, CLS3474) plates at a density of 1000 cells/well. After 16–18 h, Day 0 assays were undertaken and the cells were treated either with vehicle control (DMSO) or RA306 at varying concentrations. After 120 h, assays were performed on replicates of vehicle control and RA306-treated cells.

## Transwell migration/invasion assay

Transwell migration and invasion assays were essentially as previously described[54]. Transwell migration chambers with 8 μm pores were purchased from Corning (cat. CLS3464-48EA) and MERCK Millipore (cat. MCEP24H48). Matrigel invasion chambers pre-coated with Matrigel were purchased from Corning (cat. 354483). Prior to seeding, cells were treated with mitomycin C at 10 μg/ml for 1 h to prevent cell division.

## In vitro kinase assay

HEK293T cells were transiently transfected with plasmids encoding wildtype PEAK1 or the CAMK2-interaction motif mutants. Cell lysates were prepared using RIPA buffer[41] supplemented with 12.5 mM β-glycerol phosphate. Cell lysates were then incubated with anti-FLAG M2 affinity-agarose beads (Sigma, cat. A2220) at 4 °C for 2 h with end-to-end rotation. After the incubation, the beads were washed extensively with ice-cold RIPA buffer and kinase reaction buffer [Tris (pH 7.5) 40 mM, $MgCl_2$ 10 mM, $CaCl_2$ 0.5 mM]. After the last wash, beads were incubated in kinase reaction buffer supplemented with 100 ng recombinant CAMK2D (Invitrogen, cat. PV3373), 100 μM recombinant calmodulin (Enzo, cat. BML-SE325-0001) and 2 mM Adenosine 5′-triphosphate (ATP) disodium salt hydrate (Sigma-Aldrich, cat. A1852-1VL) for 30 min at 37 °C. Following incubation, the immune complexes were eluted with SDS-PAGE loading buffer and subjected to Western blotting analysis.

## Animal studies

All procedures involving mice were conducted in accordance with NHMRC regulations on the use and care of experimental animals and the study protocol approved by the Monash University Animal Ethics Committee (Ethics numbers: 23095, 30360). The maximal tumour size/burden permitted was 1000 mm³ or 10% body weight and this burden was not exceeded. Power calculations performed indicated group sizes of $n = 8$ per treatment group were required to detect a significant difference in tumour growth inhibition of 50% between treated and control (significance level: 0.05; power: 90%). All in vivo experiments were performed with 6-week-old female BALB/c athymic nude mice obtained from the Animal Resources Centre (Canning Vale, Australia) and housed in sterile conditions at Monash Animal Research Platform. This facility uses a 12–12 h light-dark cycle, 22–24 °C. Female mice were utilised due to the requirement for tumour growth in the mammary fat pad. Throughout the study, each cage was randomly assigned to the experimental groups without considering any other variable. Mice were randomly divided into four equal groups, cas9-control cells with captisol (control), cas9-control cells with RA306, cas9-sgPEAK1 cells with captisol and cas9-sgPEAK1 cells with RA306.

## Xenograft model

Mice ($n = 40$) were injected with $2 \times 10^5$ MDA-MB-231_HM Cas9 control or sgPEAK1 stable knockdown cells suspended in 20 μl PBS into the fourth mammary fat pad. One day post-injection, RA306 (30 mg/kg) or captisol-only control was administered via daily oral gavage until the end of the experiment. To measure tumour burden, mice were injected with luciferin (150 mg/kg) by an intraperitoneal route. Tumour burden was measured weekly using the AMI-HTX imaging system (Spectral Instruments Imaging). Results are presented as mean +/− SEM of total BLI signal (photons/s).

## Tumour metastasis model

Mice ($n = 40$) were injected with $2 \times 10^5$ MDA-MB-231_HM Cas9 control or sgPEAK1 stable knockdown cells suspended in 20 μl PBS into the fourth mammary fat pad. One day post-injection, RA306 (30 mg/kg) or captisol-only control was administered via daily oral gavage until the end of the experiment. When the primary tumours from each group reached ∼100 mm³ in size, they were resected. Tumour metastasis to secondary sites was then measured weekly using the AMI-HTX imaging system (Spectral Instruments Imaging). Organs were dissected immediately following the final BLI image and re-imaged in the AMI-HTX ex vivo to allow for direct measurement of organ-specific tumour burden via BLI. Results are presented as mean tumour luciferase intensities +/− SEM of total BLI signal (photons/s).

## Tumour tissue homogenisation

Following dissection at the experimental endpoint, tumour tissues were transferred immediately to dry ice. Tumour tissues were cut into small pieces and added to a 1.5 ml tube containing RIPA buffer and Zirconia tissue disruption beads (Research Products International, cat. 9835) then homogenised for 10 s at low speed and 30 s at high speed at room temperature.

To prepare samples for Western blotting, homogenised samples were centrifuged for 20 min at $20,000 \times g$ at 4 °C. The supernatant was then collected, and the protein concentration determined using either Bradford Ultra Protein Assay Reagent (Expedeon, cat. EP-BFU05L) or Pierce TM BCA Protein Assay Reagents (Thermo Scientific, cat. 23225). The protein concentration of each sample was subsequently adjusted using RIPA buffer, and SDS-PAGE loading buffer was added prior to Western blotting analysis.

## Association of PEAK1 and CAMK2D/G with overall survival and distant metastasis-free survival

For overall survival analysis, publicly-available mRNA expression, mutation profile and associated overall survival data from 2509 breast cancer patients as part of the METABRIC trial[55] were downloaded from the cBioPortal for Cancer Genomics portal (https://www.cbioportal.

org/). Breast cancer patients were divided and classified into three main groups, each having either low, or medium, or high expression of *CAMK2D, CAMK2G* or *PEAK1* individually. Main groups were then further categorised into four sub-groups exhibiting different combined expression patterns: (*i*) *CAMK2D* and *PEAK1* both high, (*ii*) *CAMK2D* and *PEAK1* both low, (*iii*) *CAMK2G* and *PEAK1* both high, (*iv*) *CAMK2G* and *PEAK1* both low, (*v*) *CAMK2D* low and *PEAK1* high, (*vi*) *CAMK2D* high and *PEAK1* low. Survival analyses comparing overall survival between these sub-groups were subsequently performed using a Log-rank test (with $p < 0.05$ considered significant). The Log-rank test statistics and survival curves were generated using the Kaplan-Meier estimate and implemented using the Logrank package[55] in MATLAB 2023a.

To determine the association with distant metastasis-free survival (DMFS), mRNA expression and DMFS data were obtained from the Gene Expression Omnibus (GEO) database as indicated under 'Data availability'. CAMK2D data were unavailable in most datasets; therefore, this analysis focused on PEAK1 and CAMK2G. To create a more robust and larger meta-dataset, we applied z-score standardisation to the gene expression data of individual cohorts and then combined all cohort data to generate a 527-patient meta-cohort. Subsequently, breast cancer patients were divided into low or high-expression groups for PEAK1 and CAMK2G, individually, based on the 33% percentile of gene expression (bottom 33% vs top 67%). Finally, patient groups were categorised into two subgroups based on combined expression patterns: (*i*) PEAK1 and CAMK2G both low, and (*ii*) PEAK1 and CAMK2G both high. The DMFS was compared between these two subgroups using a log-rank test ($p < 0.05$ considered significant). Log-rank test statistics and survival curves were generated using the *survival* (version 3.5.8) and *survminer* (version 0.4.9) packages in R.

### Statistics and reproducibility
Experimental data were subject to appropriate statistical analyses (two-tailed unpaired *t* tests, 1 or 2-way ANOVA with appropriate post-hoc tests, Chi-square analysis) as detailed in the corresponding figure legends. Power calculations were utilised to determine the sample sizes for animal work, as detailed under 'Animal studies'. Animal cages were randomly assigned to the experimental groups. All quantified cell-based experiments involved the use of at least $n = 3$ independent experiments unless specified. No data were excluded. The Investigators were not blinded to allocation during experiments and outcome assessment.

### Reporting summary
Further information on research design is available in the Nature Portfolio Reporting Summary linked to this article.

### Data availability
The mass spectrometry proteomics data for the spectral library and DIA analysis have been deposited to the ProteomeXchange Consortium via the PRIDE[56] partner repository with the dataset identifier PXD044872. Project Webpage: http://www.ebi.ac.uk/pride/archive/projects/PXD044872. FTP Download: https://ftp.pride.ebi.ac.uk/pride/data/archive/2025/01/PXD044872. The gene expression data used in this study are available in the Gene Expression Omnibus (GEO) database under the following accession codes: Princess Margaret Cancer Centre (database accessions GSE6532, https://www.ncbi.nlm.nih.gov/geo/query/acc.cgi?acc=gse6532 and GSE9195, https://www.ncbi.nlm.nih.gov/geo/query/acc.cgi?acc=GSE9195), Veridex LLC (GSE12093, https://www.ncbi.nlm.nih.gov/geo/query/acc.cgi?acc=gse12093) and Bayer Technology Services GmbH (GSE11121, https://www.ncbi.nlm.nih.gov/geo/query/acc.cgi?acc=gse11121). All other data supporting the findings in this work are included in the main article, supplementary information or source data file. Source data are provided in this paper.

### Code availability
AlphaFold models and associated metadata to accompany this paper have been deposited in the Zenodo database and are available at https://zenodo.org/records/14715248.

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

## Acknowledgements

The authors acknowledge the Monash Proteomics and Metabolomics Facility, Monash University, for the provision of MS instrumentation, training and technical support, and Monash Micro Imaging for assis-tance with the calcium imaging experiments. The work was supported by research grants to R.J.D. from the National Health and Medical Research Council of Australia (Ideas Grant 2003599) and Australian Research Council (ARC) (DP220103638 and DP190103672) and to MMS (NIH R35GM145376). D.R.C. acknowledges fellowship support from the National Breast Cancer Foundation (NBCF Fellowship IIRS-20-032), A.M.E. is a Victorian Department of Health and Human Services Victorian Cancer Agency Mid-Career Fellow (MCRF21036) and C.B. is an ARC DECRA Fellow (DE240100992). VCBN is a member of the Molecular and Cellular Biology Graduate Programme, University of Massachusetts, Amherst, MA 01003, USA and is supported by a fellowship from the

University of Massachusetts as part of the Chemistry-Biology Interface Training Programme (National Research Service Award T32 GM139789).

## Author contributions

Conceptualisation, R.J.D.; Methodology, X.Y., X.M., D.R.C., C.B., S.B., S.S., L.K.N., A.C.C., M.M.S., A.M.E. and R.J.D.; Investigation, X.Y., X.M., D.R.C., E.V.N., K.C.C., C.H., S.L.L., T.Z., C.B., B.V.N., S.B., S.S. and T.C.C.L.K.S.; Writing – Original Draft, X.Y., X.M. and R.J.D.; Writing – Review and Editing, T.R.C., M.M.S., A.M.E. and R.J.D.; Visualisation, X.Y., T.R.C. and R.J.D.; Funding acquisition, D.R.C., M.M.S., C.B., A.M.E. and R.J.D.; Supervision, X.M. and R.J.D.; Project administration, R.J.D.

## Competing interests

The authors declare no competing interests.
