## [Transparent Peer Review file · Nature Communications]

Activation of CAMK2 by pseudokinase PEAK1 represents a targetable pathway in triple negative breast cancer

Corresponding Author: Professor Roger Daly

Version 0:

Reviewer comments:

Reviewer #1

(Remarks to the Author)

General comments:

In this manuscript, the authors illustrated a novel mechanism through which PEAK1 triggered Ca²⁺ increase, which led to the activation of CAMK2. The activated CAMK2 would phosphorylate PEAK1, which might be involved in cancer metastasis. The xenograft model revealed that knockout of PEAK1 and RA306, a CAMK2 inhibitor, could reduce tumour size and metastasis.

Regarding the signaling cascade, PEAK1 would employ PLCr1 to enhance the intracellular Ca²⁺ level to activate CAMK. In this stage, the author did not explain if monomer or dimer contributed to the recruitment of SFK and PLCr1. Based on the information, CAMK2 would bind to the homodimer of PEAK1 and mediate phosphorylation, which is essential for PEAK1 to modulate metastasis. However, the author did not provide information on the downstream mechanism through which PEAK1 homodimer enhances metastasis. It has been demonstrated that CAMK2 would be involved in the assembly of actin cytoskeleton, which would affect cell migration (Cancer Res. 2023 Sep 1;83(17):2889-2907. doi: 10.1158/0008-5472.CAN-22-1622. PMID: 37335130). In the current manuscript, it is not surprising that RA306 could suppress metastasis as it can inhibit cell migration by interfering with actin assembly. Hence, the contribution of PEAK1 to metastasis still remains to be addressed. Nevertheless, the current study does indeed demonstrate that using RA306 could suppress metastasis, which has translational value.

The authors should clarify the importance of PEAK1 phosphorylation mediated by CAMK2D or CAMK2G on metastasis. This information is lacking in the current manuscript. The roles of CAMK2 on metastasis have been documented, but the role of PEAK1 in TNBC is unclear. It has been shown that PEAK1 could promote EMT in lung cancer (Cell Death Dis. 2018 Jul 23;9(8):802. doi: 10.1038/s41419-018-0817-1. PMID: 30038287; PMCID: PMC6056550.). The authors might refer to this study to enrich the current manuscript.

I would recommend that the authors make a major revision.

Specific comments that should be addressed:

- In Fig 1, the authors illustrated how they identified CAMK2D and CAMK2G using MS. Next, they employed PLA assay to study the interaction. They used siRNA to reduce the expression of PEAK1 and PEAK2. The results showed that in the siPEAK1 or siPEAK2 treated cells, the interaction with CAMK2D and CAMK2G was reduced significantly. How about the results of PEAK2/CAMK2D, PEAK2/CAMK2G in siPEAK1 treated MDA-MB-231? Similarly, how about the results of PEAK1/CAMK2D, PEAK1/CAMK2G in the siPEAK2 treated cells? The interaction could involve PEAK1/2 homodimer and PEAK1/2 heterodimer, but the results of Fig 1E and Fig 1F solely indicate that of knockdown of PEAK1 (Fig 1E) and PEAK2 (Fig 1F) PEAK/CAMK interaction leading to a signal reduction in PLA. Conclusions drawn from the given results are compromised. Moreover, CRISPR/Cas9-mediated knockout cell lines should be used. For example, for PEAK1 KO cells, PLA assay will only detect the interaction between PEAK2 homodimer and CAMK2D and CAMK2G. The results will be more clear.

- In Fig. 2E-F, PEAK1 knockdown or PEAK1 knockout should be employed to verify the effect of PEAK1 on CAMK2D/G inactivation and Ca²⁺ change in the BAPTA-AM-treated cells.

- In extended data Fig 3, the authors demonstrated that inhibition of Src family kinase by eCF506 could compromise the effect of PEAK1 on PLCr1 phosphorylation. Which form of PEAK1 is involved in this activation step - PEAK1 monomer, PEAK1/1 homodimer or PEAK1/2 heterodimer?
- In line 175-176 is it mentioned "...the Nterm region alone did not associate with CAMK2D and G, indicating that (Fig 3B)". However, CAMK2G is absent from the blots for Fig 3B and 3D. The results of PEAK2 deletion mutants are presented later in extended data Figure 4, but no data for CAMK2G interaction is shown at all.
- In Fig. 3E, the authors showed that deletion of 301-324 could reduce the interaction with CAMK2G. How about the effect of deletion of 261-300 on the interaction? One follow-up question, could PEAK1 del301-324 interact with wild-type PEAK1 to form a dimer?
- In Fig. 3F-G, the authors showed the PEAK1 mutant could abolish the CAMK2D/G activation. How about the effect of the PEAK1 mutant on Ca²⁺ level and PLCr1 phosphorylation? Also, in Fig 3F, the western blot results have some unnatural lines between the lanes. Please see the PowerPoint file for details. The authors should address the issue.
- Line 238 mentions R297 (-9) of PEAK1. However in figure 4A, R297 is under the column (-8), although in figure 4E, the canonical numbering -9 is seen at the branch leading to R297. It needs to be clarified whether R297 is (-9) or (-8).
- In line 241, it is mentioned the predicted formatin of salt bridges by PEAK1 D308, D310 and D311 across the surface of CAMK2 (Figure 4F). However D311 is not shown in Fig 4F. Moreover, the figure legend for Fig 4E-F does not explain clearly how to interpret the figures. Do the dotted lines represent the salt bridges?
- In line 243, extended data Fig. 6 showed the results from alphaFold of CAMK and PEAK 291-320 interaction. The modelling was undertaken with CAMK2A (line 241-242). Should not similarity or identification analyses be done between the CAMK2 isoforms? Although it is mentioned that the binding site is conserved across all CAMK2 isoforms, there is no reference or data to support this. The proper way should be doing the same analyses using CAMK2D and CAMK2G rather than CAMK2A.
- In Fig. 5A, how do RLR-AAA and RR-EE mutants affect PLCr1 interaction and phosphorylation?
- To supplement Fig. 6C, an in vitro kinase assay should be performed using PEAK1 peptide, as mentioned in the materials and methods, to confirm that CAMK2D can phosphorylate the serine (S300; coordinate referred to Fig. 4A) in RNKS of PEAK1.
- To further support extended data Fig. 7C, PEAK S300A and PEAK S300D mutants should be cloned and study if the phosphomimic (S300D) will affect the interaction with 14-3-3 compared to S300A and wild type.
- To further support extended data Fig. 8 and main Fig.7, the correlation between high PEAK1 and CAMK2G/D and metastasis should be determined to highlight their clinical significance.
- For Fig. 7, before jumping to migration assay, the authors should determine if RA306 and PEAK1 knockout would affect i) cell viability by MTT assay and apoptotic assay, ii) anchoring independence by Soft agar colony formation assay and iii) wound healing assay to address which aspect PEAK1 and CAMK2 affect metastasis. If the treatment of RA306 or PEAK knockout could induce apoptosis, then the author would not be detecting migrated and metastatic cells because the cancer cells would already have been killed.
- In Fig. 7F, the primary tumour was missing in the control. In the figure legend, the author claimed the primary tumour was resected. However, no wound or resection site on the mice can be seen in the image. The authors might need to provide additional evidence.

Reviewer #2

(Remarks to the Author)

"I co-reviewed this manuscript with one of the reviewers who provided the listed reports. This is part of the Nature Communications initiative to facilitate training in peer review and to provide appropriate recognition for Early Career Researchers who co-review manuscripts."

Reviewer #3

(Remarks to the Author)

In this manuscript, Yang et. al identified the association of PEAK1 and CAMK2 and feed-forward mechanism involving PEAK1/PLCg1/Ca²⁺ signaling. They also proved that CAMK2 inhibitor could attenuated PEAK1-dependent tumor growth, migration and invasion, which means CAMK2 as a therapeutically target in TNBC. Overall, the findings will arouse the interest of the researchers in the related areas. However, the current version of the material or figures is still inadequate for publication.

1. The raw mass spectrometry data is currently unavailable. The authors should make these data publicly accessible.

2. It is confused that the authors utilized MaxQuant (version 1.5.2.8) for the analysis of the raw files generated by DIA analysis.
3. The manuscript lacks detailed methods regarding LC-MS/MS experiments. Parameters for LC separation and MS detection should be included in the methods section.
4. More stringent data quality control measures for the MS experiments should be incorporated into the supplementary data.
5. Numerous candidate interacting proteins were identified through affinity purification coupled with tandem mass spectrometry experiments, but only several proteins are discussed and focused in Fig 1A-C.
6. Why you focus on PEAK1 rather than PEAK2? What's the difference between them? You have demonstrated that CAMK2 interact with both PEAK1 and PEAK2.
7. I suggested enhancing the aesthetics of the volcano map displayed in Figure 1. The color is not favorable. Additionally, the font size of the axis labels should be adjusted to ensure they are not too small compared to other text elements. Similar improvements should be made to the other figures as well, as the current presentation is not satisfactory for publication.
8. Most of WB figures in your manuscript do not have reference strips (such as GAPDH, Actin, etc.). Why not use these? Are you regarding 14-3-3 as reference? If so, why you use 14-3-3 as a reference?
9. In line 234, the first appearance of KD requires detailed definition.
10. The resolution of the Extended Figure 5 and 8 needs to be improved.
11. You explored the effect of knockdown of either CAMK2D or G on the PEAK1-dependent cell migration and invasion. Therefore, in Figure 7B, the p-value of the difference between Vector and PEAK1 in si-CAMK2D or G should be marked.
12. Since RA306 and gene knock-out have the same effect on the impairment of tumor growth, which of these two targets is more likely to be applied in clinic? Is there any clinical research basis on these two targets?

Reviewer #4

(Remarks to the Author)

I co-reviewed this manuscript with one of the reviewers who provided the listed reports.

Reviewer #5

(Remarks to the Author)

The PEAK family proteins PEAK1 and PEAK2 are human pseudokinases with a similarity to the GSK family of protein kinases. However, due to the lack of the critical kinase motifs DFG and GXGXXG do not exhibit any kinase activity. Still, these proteins are involved in regulating several cellular processes, probably by acting as scaffolds in some cellular signaling pathways. Interestingly, these proteins were shown to act as oncogenes in some cancers, including triple negative breast cancer. Therefore, reduction of their activity/function may be important in combating these cancers, requiring a better understanding of their mechanisms of action. Here, the authors identified CAMK2D and CAMK2G as effectors of PEAK1, acting via a novel feed-forward mechanism including CAMK2 phosphorylation of PEAK1 to enhance association with PEAK2, which is critical for PEAK1 oncogenic signaling. In view of these findings, the authors used the CAMK2 inhibitor RA306 and showed that it inhibits PEAK1-enhanced migration and invasion in vitro, attenuated growth and blocked metastasis in TNBC xenografts.

Overall, this is an important study that identifies the mechanism of action of PEAK1 and establishes it as a novel regulator of Ca²⁺ and certain PTKs signaling. In addition, the study offers a new strategy to combat PEAK1-overexpressing triple negative breast cancer. Although the study is generally well-written and the results are believable (except of Fig. 1 E&F and Fig. 3F), I do have some comments that should be addressed to allow publication. These are as follows:

- 1) Fig. 1 E&F is problematic. In both, the lower panels (CAMK2G) are smeared, and the number of dots seems to be much below the numbers that appear in the quantification. Another point is the lack of signals of CAMK2G/PEAKs in the nucleus, which is unlike the nuclear signals of CAMK2D/PEAKs. Since CAMK2G does not localize to the nucleus (at least in some cells) the reason for the lack of PLA signal is not clear and should be explained.
- 2) In Fig. 2, what are the units of the relative p-CAMK2 level. The units are missing in other figures as well.
- 3) In many figures it is suggested to move the result panel to the top of the figure (e.g. in Fig. 2A and 2C move the second panel (CAMK-pT287) to the top), because these are actually the important parts of the figures.
- 4) In Fig. 3D, what are the two bands seen in the flag blot of del 261-300 and 261-324?
- 5) In Fig. 3E, why is the deletion marked "d" and not del? The MW size of d1-260 seems to be too high.
- 6) Fig. 3F is problematic and should be replaced. It seems that the Del 301-324 is taken from another blot, but this is not properly marked or explained. It is important to show the results on one blot for a better side by side comparison. In addition, the band in PEAK1-vector is strange. Please check.
- 7) In the extended data figure 2A, there is something wrong in the top legends.

7) In extended data figure 4, why is the deletion marked and not del or d as in other figures?

8) The term SFK is not spelled out anywhere in the text. Can the authors provide information on the relevant Src family kinases involved?

9) In several figures, the authors normalized the tested parameters by dividing the intensity of the tested parameter by the intensity of the total band. This is problematic because the different antibodies may have a different recognition linearity (namely, the change in intensity by each antibody does not necessarily reflect the difference in the amount of proteins / phosphorylation). It is suggested to use a calibration curve for each of the antibodies to correlate the intensity to the protein amounts.

10) Although this point does not affect the results, in my mind the addition of 12.5 mM beta glycerophosphate to RIPA is not necessary, as the other components of the buffer are sufficient to inhibit phosphatases.

Version 1:

Reviewer comments:

Reviewer #1

(Remarks to the Author)

The authors have adequately answered all issues raised. There is only one comment arising for consideration.

Based on the study, high expression of PEAK1 would enhance Ca²⁺ level. Once PEAK1 is overexpressed, it elevates intracellular Ca²⁺ immediately, activating Ca²⁺ signalling. Prolong activation of Ca²⁺ signalling is apoptotic. The revised Fig2G, H, only show single time point experimental result. We cannot tell from this whether there was a negative feedback mechanism to reduce Ca²⁺ in the later time point. The original manuscript, Figure 2G and H did indeed present sequential data showing this phenomenon. The authors might consider including the sequential data in the supplementary figures to exclude the possibility that the PEAK1 interactome could be involved into anti-apoptotic pathways.

Reviewer #2

(Remarks to the Author)

Reviewer #3

(Remarks to the Author)

The authors have addressed our concerns.

Reviewer #4

(Remarks to the Author)

Reviewer #5

(Remarks to the Author)

The authors successfully answered all my concerns. I have no further comments.

Re: Nature Communications manuscript NCOMMS-24-14332-T

'Feed-forward stimulation of CAMK2 by the oncogenic pseudokinase PEAK1 generates a therapeutically 'actionable' signalling axis in triple negative breast cancer' by Yang *et al*

Dear Reviewers,

We have undertaken a significant volume of additional experimentation and analysis in order to address issues raised and have revised the manuscript accordingly. In particular, we have:

- clarified how the PEAK1/CAMK2 axis regulates tumour growth and metastasis through additional work characterizing the biological effects of RA306 and PEAK1 gene knock-out, including their impact on apoptosis, together with interrogation of publically-available patient data and consideration of the associated literature
- confirmed that PEAK1 mutants impaired in direct CAMK2 binding retain the ability to enhance intracellular calcium and PLC γ 1 phosphorylation, supporting our feed-forward model of CAMK2 regulation
- provided important source data, methodological detail and clarification, including for the MS experimentation, metastasis model and Western blotting
- Improved data presentation and explanation, including for the AlphaFold modelling.

Detailed responses are provided below.

REVIEWER COMMENTS

Reviewer #1 (TNBC therapy):

General comments:

In this manuscript, the authors illustrated a novel mechanism through which PEAK1 triggered Ca²⁺ increase, which led to the activation of CAMK2. The activated CAMK2 would phosphorylate PEAK1, which might be involved in cancer metastasis. The xenograft model revealed that knockout of PEAK1 and RA306, a CAMK2 inhibitor, could reduce tumour size and metastasis.

Regarding the signaling cascade, PEAK1 would employ PLC γ 1 to enhance the intracellular Ca²⁺ level to activate CAMK. In this stage, the author did not explain if monomer or dimer contributed to the recruitment of SFK and PLC γ 1.

Work from the Roche laboratory has established that dimerization of PEAK2 and PEAK3 is critical for activation of the associated tyrosine kinases CSK and PYK2, respectively (Lecointre *et al*, *Structure*, 26, 545, 2018; Ounoughene *et al*, *Cancers* 13, 6344, 2021). However, interrogation of our BiCAP proteomics dataset did not reveal association of PEAK1, PEAK2 or the PEAK1/2 heterodimer with any SFK members, or PLC γ 1. Consequently, signalling from PEAK1 through SFKs to PLC γ 1 appears to be an indirect mechanism that does not involve direct *recruitment* of SFKs or PLC γ 1 to PEAK1. We believe that definition of this mechanism is beyond the scope of the manuscript but have described the lack of detectable association of SFKs and PLC γ 1 with PEAK1/2 in the Results of the revised manuscript (**Page 8**).

Based on the information, CAMK2 would bind to the homodimer of PEA1 and mediate phosphorylation, which is essential for PEA1 to modulate metastasis. However, the author did not provide information on the downstream mechanism through which PEA1 homodimer enhances metastasis. It has been demonstrated that CAMK2 would be involved in the assembly of actin cytoskeleton, which would affect cell migration (Cancer Res. 2023 Sep 1;83(17):2889-2907. doi: 10.1158/0008-5472.CAN-22-1622. PMID: 37335130). In the current manuscript, it is not surprising that RA306 could suppress metastasis as it can inhibit cell migration by interfering with actin assembly. Hence, the contribution of PEA1 to metastasis still remains to be addressed. Nevertheless, the current study does indeed demonstrate that using RA306 could suppress metastasis, which has translational value.

The authors should clarify the importance of PEA1 phosphorylation mediated by CAMK2D or CAMK2G on metastasis. This information is lacking in the current manuscript. The roles of CAMK2 on metastasis have been documented, but the role of PEA1 in TNBC is unclear. It has been shown that PEA1 could promote EMT in lung cancer (Cell Death Dis. 2018 Jul 23;9(8):802. doi: 10.1038/s41419-018-0817-1. PMID: 30038287; PMCID: PMC6056550.). The authors might refer to this study to enrich the current manuscript.

The focus of the Cancer Research paper cited by the reviewer is not CAMK2, it is calcium/calmodulin-dependent protein kinase kinase 2, CAMKK2. **This is a completely different kinase that does not signal in the same manner as CAMK2.** Therefore the reviewer's comment that the effect of RA306 is 'not surprising' is unjustified.

With regard to clarifying 'the importance of PEA1 phosphorylation mediated by CAMK2D or CAMK2G on metastasis' we agree that the mechanism of action of the PEA1/CAMK2 axis requires further elaboration. In brief, one mechanism is provided by our demonstration that CAMK2-mediated phosphorylation of PEA1 promotes heterodimerization of PEA1 with PEA2 (Fig 6D). This is because heterotypic association of PEA1 and PEA2 is required for PEA1 to promote cell migration, as described in our previous work (Liu *et al*, J Biol Chem 291, 21571, 2016). The underlying mechanism is that PEA2 recruits/activates critical effectors such as Csk and Stat3. Also, the activated CAMK2 may phosphorylate other downstream targets that promote metastasis, such as the Rho family GEF Tiam1 (Saneyoshi *et al*, Neuron 102, 1199-1210, 2019), and regulate the focal adhesion proteins FAK and paxillin (Easley *et al*, Cell Motil Cytoskeleton 65, 662-674, 2008). We have added text describing these mechanisms to the revised manuscript (**Discussion, Pages 19-20**). As highlighted later in the rebuttal, we also discuss how the PEA1/CAMK2 signalling axis contributes to tumour growth and metastasis through effects on cellular processes such as proliferation, migration and invasion on **Page 15**.

The comment 'The roles of CAMK2 on metastasis have been documented' is an overstatement. We assume that the reviewer is referring to the aforementioned Cancer Res paper on the different kinase CAMKK2. In fact, there is very limited direct evidence for CAMK2 being required for metastatic progression. In part, this is because until recently, pharmacological targeting of CAMK2 was reliant on 'tool' compounds such as KN93 which demonstrate many 'off-target' effects. We have identified one paper focusing on prostate cancer that utilized CRISPR-mediated KO to demonstrate a role for CAMK2 in experimental metastasis (Yu *et al*, Cancer Res 78, 2490, 2018), and have cited it in the revised manuscript (**Page 19**). However our study is the first to directly demonstrate a role for CAMK2 in breast cancer metastasis.

The reviewer comments that 'the role of PEAk1 in TNBC is unclear'. However my laboratory has previously demonstrated that PEAk1 is overexpressed in TNBC where it promotes cell proliferation, migration and invasion, abnormal morphogenesis in 3D culture, and tumour growth and metastasis (Croucher *et al*, Cancer Res 73, 1969, 2013; Liu *et al*, J Biol Chem 291, 21571, 2016; Abu-Thuraia *et al*, Nature Comms 11, 1-20, 2020). These papers were cited in the original manuscript but alongside other papers focusing on additional cancers. For clarity, we have now cited these TNBC papers separately (**Page 3**). The paper on PEAk1 and EMT in lung cancer has also been cited in this section and we thank the reviewer for this suggestion.

Overall, we believe these modifications greatly improve the clarity of the manuscript regarding how PEAk1/CAMK2 regulates tumour growth and metastasis and further emphasize the novelty of our findings.

Specific comments that should be addressed:

- In Fig 1, the authors illustrated how they identified CAMK2D and CAMK2G using MS. Next, they employed PLA assay to study the interaction. They used siRNA to reduce the expression of PEAk1 and PEAk2. The results showed that in the siPEAk1 or siPEAk2 treated cells, the interaction with CAMK2D and CAMK2G was reduced significantly. How about the results of PEAk2/CAMK2D, PEAk2/CAMK2G in siPEAk1 treated MDA-MB-231? Similarly, how about the results of PEAk1/CAMK2D, PEAk1/CAMK2G in the siPEAk2 treated cells? The interaction could involve PEAk1/2 homodimer and PEAk1/2 heterodimer, but the results of Fig 1E and Fig 1F solely indicate that of knockdown of PEAk1 (Fig 1E) and PEAk2 (Fig 1F) PEAk/CAMK interaction leading to a signal reduction in PLA. Conclusions drawn from the given results are compromised. Moreover, CRISPR/Cas9-mediated knockout cell lines should be used. For example, for PEAk1 KO cells, PLA assay will only detect the interaction between PEAk2 homodimer and CAMK2D and CAMK2G. The results will be more clear.

We respectfully disagree with the reviewer regarding these comments. The purpose of these experiments is to confirm that PEAk1 closely co-localizes with CAMK2 at endogenous expression levels, as would be expected if the two proteins interact, and the same for PEAk2 and CAMK2. The controls we have utilized are entirely appropriate for demonstrating that the PLA signal is dependent on the presence of the interacting partner in question ie PEAk1 siRNA for the PEAk1/CAMK2 interaction and PEAk2 siRNA for the PEAk2/CAMK2 interaction. In both cases, knockdown significantly reduces the PLA signal, indicating that the interacting partner is required. The conclusions drawn are **not** compromised.

The reviewer then raises an additional question of whether, for example, PEAk2 siRNA affects the PEAk1/CAMK2 interaction, since based on our MS data, the PEAk1/2 heterodimer binds CAMK2. However the results of this experiment would be difficult to interpret. While it is true that knocking down PEAk2 would free up PEAk1 from the heterodimer, this may then lead to increased formation of PEAk1 homodimers from the 'released' PEAk1, and these would then bind CAMK2 and be detected by PLA. In addition, knocking down PEAk2 may cause signalling changes that affect the PEAk1/CAMK2 interaction. For these reasons we have kept the experiment in its simpler, easily interpretable form.

- In Fig. 2E-F, PEAk1 knockdown or PEAk1 knockout should be employed to verify the effect of PEAk1 on CAMK2D/G inactivation and Ca²⁺ change in the BAPTA-AM-treated cells.

This experiment is designed to test the requirement for intracellular Ca²⁺ in PEAk1-mediated CAMK2 activation. In control cells, PEAk1 overexpression enhances CAMK2 activation, as

determined by T286 phosphorylation. In the presence of the Ca²⁺ chelator BAPTA-AM, this effect is not observed. It is unclear why the reviewer proposes inclusion of PEAK1 knockdown/KO in this experiment. The effect of PEAK1 overexpression/knockdown on CAMK2 activation are shown in Fig 2A-D.

- In extended data Fig 3, the authors demonstrated that inhibition of Src family kinase by eCF506 could compromise the effect of PEAK1 on PLCr1 phosphorylation. Which form of PEAK1 is involved in this activation step - PEAK1 monomer, PEAK1/1 homodimer or PEAK1/2 heterodimer?

In terms of PLCγ1 and Ca²⁺ regulation by PEAK1, we have focused our effort during manuscript revision on confirming that mutation of the PEAK1 interaction motif for CAMK2 does not affect its ability to enhance PLCγ1 phosphorylation or intracellular Ca²⁺ (**Revised Fig 5**) thereby providing further support for the 'feed-forward' model presented in Fig 8. We believe that this is the critical issue absent from the original manuscript, and data from these experiments are described later in response to a separate comment. We have not specifically tested the requirement for PEAK1 dimerization in signalling through SFKs to PLCγ1, primarily because, as discussed above, interrogation of our proteomics dataset indicates that the mechanism involved is indirect ie does not involve direct association of PEAK1 with these proteins. Consequently, it is unclear what insights this additional work would bring without also characterizing this indirect signalling mechanism, which is beyond the scope of the manuscript. Also, determining whether this is the homodimer or heterodimer is challenging since it would involve reconstitution of PEAK1 wildtype and dimerization-defective mutants into PEAK1/PEAK2 double KO and PEAK1 single KO cells. Again, in the absence of a defined downstream pathway we are not convinced that this would add significantly to the manuscript.

- In line 175-176 is it mentioned “...the Nterm region alone did not associate with CAMK2D and G, indicating that (Fig 3B)”. However, CAMK2G is absent from the blots for Fig 3B and 3D. The results of PEAK2 deletion mutants are presented later in extended data Figure 4, but no data for CAMK2G interaction is shown at all.

The reviewer is correct in highlighting that association of the Nterm region alone with CAMK2G is not shown in Fig 3B. We thank the reviewer for this comment and have corrected the corresponding text (**Page 8**). However, in Fig 3E we demonstrate that like CAMK2D, deletion of PEAK1 amino acids 301-324, but not 1-260, leads to loss of CAMK2G binding. This is highlighted in the original text (**Page 9**).

- In Fig. 3E, the authors showed that deletion of 301-324 could reduce the interaction with CAMK2G. How about the effect of deletion of 261-300 on the interaction?

We haven't determined the effect of the 261-300 mutant but predict that this would lose interaction with CAMK2 because the deletion would disrupt the CAMK2 interaction motif (Fig 4A).

One follow-up question, could PEAK1 del301-324 interact with wild-type PEAK1 to form a dimer?

Yes, PEAK1 del301-324 could dimerize with endogenous wildtype PEAK1 since the SHED region responsible for dimerization is still intact in the mutant. However the association studies

in Fig 3 were undertaken in transiently transfected HEK293T cells which exhibit high levels of expression of the particular mutants. Under these conditions the mutants are in excess over the endogenous wildtype protein and so the dimers formed would be predominantly mutant homodimers, allowing the effect of the deletion on CAMK2 association to be determined.

- In Fig. 3F-G, the authors showed the PEA1 mutant could abolish the CAMK2D/G activation. How about the effect of the PEA1 mutant on Ca²⁺ level and PLCr1 phosphorylation?

We agree that the effect of mutants featuring deletion or mutation of the CAMK2-interaction motif (CIM) on intracellular Ca²⁺ is an important question. According to our model (Fig 8), deletion or mutation of the CIM should not affect the ability of PEA1 to increase intracellular Ca²⁺, as this pathway should be CIM-independent. However it is important to confirm this. Rather than use the d301-324 deletion mutant, which deletes part of the CIM, we have evaluated the effect of the RLR-AAA and RR-EE CIM mutants on intracellular Ca²⁺ as these also lose the ability to promote CAMK2 activation (**Revised Fig 5E-G**) but represent less severe structural alterations. PEA1 overexpression leads to a significant increase in intracellular Ca²⁺ at steady state (Fig 2G-H) and importantly, a significant increase is also observed for the RLR-AAA and RR-EE mutants (**Revised Fig 5A-B, Pages 11-12**), indicating that their inability to promote sustained activation of CAMK2 is not due to a failure to enhance intracellular Ca²⁺ but rather lack of CAMK2 interaction. Consistent with these data, the 2 mutants also significantly increase PLC γ 1 phosphorylation in a similar manner to wildtype PEA1 (**Revised Fig 5C-D, Pages 11-12**). We thank the reviewer for raising this issue, as the new data provide important additional evidence for our model.

- Also, in Fig 3F, the western blot results have some unnatural lines between the lanes. Please see the PowerPoint file for details. The authors should address the issue.

Reviewer Figure 1. Original images for Fig 3F.

We apologize for this suboptimal data presentation. A lane was cropped from the blot since this corresponded to an irrelevant mutant. We have indicated the cropping by a dotted line in the revised **Fig 3F** and explained this in the figure legend (**Page 42**). The uncropped images are shown in source data and in **Reviewer Figure 1 (above)**.

- Line 238 mentions R297 (-9) of PEA1. However in figure 4A, R297 is under the column (-8), although in figure 4E, the canonical numbering -9 is seen at the branch leading to R297. It needs to be clarified whether R297 is (-9) or (-8).

R297 should be -9. This has been corrected on **Revised Fig 4A**.

- In line 241, it is mentioned the predicted formatin of salt bridges by PEAK1 D308, D310 and D311 across the surface of CAMK2 (Figure 4F). However D311 is not shown in Fig 4F. Moreover, the figure legend for Fig 4E-F does not explain clearly how to interpret the figures. Do the doted lines represent the salt bridges?

We have modified the main text (**Page 11**) and labelled D311 and D315 in the figure for clarity. However the main chain of PEAK1 obstructs the visibility of D311, so the latter is indicated in the background. There are numerous electrostatic interactions across the groove of CAMK2, namely salt bridges between D314:K187, D310:R52, and D308:K56/R52. D311 forms electrostatic interactions with W170. These are all shown in **Revised Figure 4F**.

The dotted lines represent contacts, or interactions, between residues. This has been stated in the figure legend (**Pages 42-43**).

- In line 243, extended data Fig. 6 showed the results from alphaFold of CAMK and PEAK 291-320 interaction. The modelling was undertaken with CAMK2A (line 241-242). Should not similarity or identification analyses be done between the CAMK2 isoforms? Although it is mentioned that the binding site is conserved across all CAMK2 isoforms, there is no reference or data to support this. The proper way should be doing the same analyses using CAMK2D and CAMK2G rather than CAMK2A.

We agree. We have performed the identity analysis and have **revised Supp Fig 6** to clearly highlight the extensive conservation between CAMK2A, B, G, and D. Specifically, we now include **Supp Fig 6C** which shows a superposition of all four isoforms modelled by AlphaFold2. Residues that differ between isoforms (i.e. those with less than complete 100% identity) are shown. It is clear that all the residues within the region that mediate contacts with PEAK1 are conserved.

Additionally, we now provide the identity matrix (showing essentially 90% identity across all four isoforms in this domain) (**Supp Fig 6C**) and a multiple sequence alignment (**Supp Fig 6D**). Taken together, it is clear that the residues involved in the PEAK1/CAMK2 interaction are conserved across isoforms.

- In Fig. 5A, how do RLR-AAA and RR-EE mutants affect PLCr1 interaction and phosphorylation?

We have undertaken additional experimentation to address this issue, as indicated in our response to the comment on Fig. 3F-G above (**Revised Fig 5**).

- To supplement Fig. 6C, an in vitro kinase assay should be performed using PEAK1 peptide, as mentioned in the materials and methods, to confirm that CAMK2D can phosphorylate the serine (S300; coordinate referred to Fig. 4A) in RNKS of PEAK1.

We thank the reviewer for this suggestion but PEAK1 doesn't phosphorylate the bound CIM peptide on this site. As described in the manuscript (**Pages 10, 17**), like the Rac GDP/GTP exchange factor TIAM1, PEAK1 exhibits a pseudosubstrate version of the conserved CAMK2 interaction motif with A rather than S/T at position 0 (amino acid 306 in PEAK1) (Fig 4A).

Despite the resemblance to a pseudosubstrate, both PEAK1 and TIAM1 are able to promote sustained activation of CAMK2, either by locking CAMK2 in an active conformation or via allosteric activation of other subunits in the CAMK2 holoenzyme (Fig 8, see also Ozden *et al*, Cell Reports 40, 111064, 2022).

Of note, in work beyond the scope of this manuscript, we have mapped CAMK2-mediated phosphorylation sites on PEAK1 and S300 was not detected.

- To further support extended data Fig. 7C, PEAK S300A and PEAK S300D mutants should be cloned and study if the phosphomimic (S300D) will affect the interaction with 14-3-3 compared to S300A and wild type.

Please refer to the response above.

- To further support extended data Fig. 8 and main Fig.7, the correlation between high PEAK1 and CAMK2G/D and metastasis should be determined to highlight their clinical significance.

This is an excellent point. In the original manuscript we presented data indicating that combined high expression of PEAK1 and CAMK2D in breast cancer significantly associated with a worse patient survival, while a non-significant trend was observed for high expression of PEAK1 and CAMK2G. We agree that the relationship between these parameters and metastasis is an important question, however undertaking the analysis was not a trivial exercise for three reasons. First, not all publically-available breast cancer cohorts have information on distant metastasis free survival (DMFS), second, many only have data on CAMK2G and not CAMK2D due to the probe sets used, and third, the cohorts may lack sufficient power. That said, after an extensive survey of available breast cancer datasets, we were able to assemble a 'meta' cohort to undertake the analysis on PEAK1 and CAMK2G (data on CAMK2D were not available) (**Revised Methods Page 34, Results Page 13-14, Supp Fig 8D**). Importantly, this analysis revealed that high association of PEAK1 and CAMK2G is positively associated with poor DMFS, significantly strengthening the paper's findings.

- For Fig. 7, before jumping to migration assay, the authors should determine if RA306 and PEAK1 knockout would affect i) cell viability by MTT assay and apoptotic assay, ii) anchoring independence by Soft agar colony formation assay and iii) wound healing assay to address which aspect PEAK1 and CAMK2 affect metastasis. If the treatment of RA306 or PEAK knockout could induce apoptosis, then the author would not be detecting migrated and metastatic cells because the cancer cells would already have been killed.

We have previously reported that CRISPR-mediated PEAK1 KO in MDA-MB-231 cells significantly reduces cell migration in a wound healing assay, cell invasion in a Boyden chamber assay, anchorage-dependent proliferation in an MTT assay and tumoursphere formation in low attachment plates (Abu-Thuraia *et al*, Nature Comms 11, 1-20, 2020). In addition, characterization of the KO cells in orthotopic xenograft and metastasis models revealed significantly reduced tumour growth and a decreased ability to form lung metastases. Taking data from the *in vitro* assays into account, it seems likely that effects on cell proliferation and migration/invasion contribute to the observed effects of the PEAK1 KO in the mouse models.

When considering our data, it is first critical to understand that the metastasis experiment was undertaken using a primary tumour resection model. Here, all tumours are grown to the same size ($\sim 100 \text{ mm}^3$) and then resected. Metastatic growth to secondary sites (that must have been initiated prior to resection) is then monitored. This allows uncoupling of effects on metastatic

growth from those on primary tumour size. Given this experimental design, it seems unlikely that greatly reduced lung metastatic growth (Fig 7F) reflects massive apoptosis in the primary tumour. To support this, we have undertaken Western blot assays for cleaved PARP (indicative of apoptosis) in cultured PEAK1 WT and KO cells, +/- RA306. No increase was observed in either setting upon PEAK1 KO and/or treatment with RA306, indicating that these treatments do not lead to enhanced apoptosis (**New Supp Fig 9D, Page 14**).

We do agree that it is important to complement our previously published assays on PEAK1 KO cells (Abu-Thuraia *et al*, Nature Comms 11, 1-20, 2020) with corresponding characterization of RA306 treatment, given that this is a new inhibitor that has not been evaluated on cancer cells previously. Consequently we have undertaken additional work demonstrating that RA306, like PEAK1 KO, significantly reduces cell proliferation under anchorage-dependent and -independent conditions (**New Supp Fig 9A-B, Page 14**). The latter assay was undertaken using a low attachment plate format that strongly correlates with soft agar growth assays (Rotem *et al*, PNAS, 112, 5708, 2015). In addition, low concentrations of RA306 that reduce cell numbers under anchorage-independent conditions do not enhance apoptosis (**New Supp Fig 9C**). We have not undertaken wound healing assays because we already provide data for transwell migration and invasion assays (Fig 7 and Supp Fig 10).

Overall, our previously published and latest data indicate that PEAK1 KO and RA306 treatment both reduce cell proliferation and cell migration and invasion, without enhancing apoptosis. The proliferative effects explain the impact on primary tumour growth (Fig 7E) while the absence of lung metastases following tumour resection cannot be explained by the reduced primary tumour size and could reflect reduced cell migratory and invasive potential and/or reduced cell proliferation at the secondary site. These possibilities are discussed on **Page 15** of the revised manuscript.

- In Fig. 7F, the primary tumour was missing in the control. In the figure legend, the author claimed the primary tumour was resected. However, no wound or resection site on the mice can be seen in the image. The authors might need to provide additional evidence.

As described in the original manuscript, this is a primary tumour resection model, so the primary tumour should be absent. The experiment was conducted by injecting MD-MBA-231 tumour cells directly into the mammary fat pad. Tumour growth was allowed to establish until tumours reached approximately 100mm³ and at this time mice were also imaged for bioluminescent signal using the AMI-HTX *in vivo* imaging system. Tumours were subsequently surgically resected in all 4 treatment groups to allow for comparison of metastatic tumour outgrowth. During resection, wounds were closed with automatic Michel clip/s. Clips were removed 2-weeks following surgery. The study endpoint was >8-weeks following tumour resection, and the resection wounds were fully healed by this time. While the wound was fully healed, evidence of the use of wound clips could be observed at study endpoint, as shown by the small lumps that are the result of skin pinched at either end of the resection site during wound closure (**Reviewer Figure 2, this rebuttal**). This pinching is only observed at the site of the fourth mammary fat pad on the right-hand side of the animal where tumour resection occurred.

Reviewer Figure 2. End of study BLI images and photos of mice 8-weeks post-tumour resection. Representative whole body and primary tumour site images of mice that have undergone tumour resection. Red arrows indicate tissue pinching and disruption due to use of wound clips post-resection.

To further clarify the study design, we have provided an additional Figure panel (**Revised Supp Fig 11D**) that presents the primary tumour BLI signals, confirming that all treatment groups exhibit the same approximate tumour size at the time of resection.

Reviewer #2 (co-reviewed with Reviewer #1):

"I co-reviewed this manuscript with one of the reviewers who provided the listed reports. This is part of the Nature Communications initiative to facilitate training in peer review and to provide appropriate recognition for Early Career Researchers who co-review manuscripts."

Reviewer #3 (mass spectrometry analysis):

This reviewer indicated that 'Yang et. al identified the association of PEAK1 and CAMK2 and feed-forward mechanism involving PEAK1/PLCg1/Ca2+ signaling. They also proved that CAMK2 inhibitor could attenuated PEAK1-dependent tumor growth, migration and invasion, which means CAMK2 as a therapeutically target in TNBC.

Overall, the findings will arouse the interest of the researchers in the related areas.'

However, the reviewer requested revision of the material and figures according to the following comments:

1. The raw mass spectrometry data is currently unavailable. The authors should make these data publicly accessible.

We apologize for this misunderstanding. The reviewer access code was in fact provided in the Nature Portfolio Reporting Summary -

"The mass spectrometry proteomics data have been deposited to the ProteomeXchange Consortium via the PRIDE [1] partner repository with the dataset identifier PXD044872".

Reviewer account details:

Username: reviewer_pxd044872@ebi.ac.uk

Password: LHFHdDGH

We believed this Reporting Summary would be provided to the reviewers and once we received the manuscript reviews and realized that it had not taken place, requested that the reviewers be sent the access code.

This text has now been added to the Data Availability Statement in the manuscript (**Page 34-35**).

2. It is confused that the authors utilized MaxQuant (version 1.5.2.8) for the analysis of the raw files generated by DIA analysis.

We have revised the Methods section in order to clarify this issue. We first generated a spectral library by DDA and then characterized the individual interactomes by DIA. We utilized this procedure because DIA is more sensitive and significantly reduces the number of missing values, greatly assisting data analysis. MaxQuant was utilized to analyse the DDA dataset. We have broken down the methodology into separate subsections for clarity and describe the procedure on **Pages 25-26**.

3. The manuscript lacks detailed methods regarding LC-MS/MS experiments. Parameters for LC separation and MS detection should be included in the methods section.

We have provided these details in the revised Methods section, **Pages 24-26**.

4. More stringent data quality control measures for the MS experiments should be incorporated into the supplementary data.

We have provided a sample correlation matrix and mean correlation scores for independent biological replicates, demonstrating high reproducibility for the approach (**New Supp Fig 1B, Page 5**).

5. Numerous candidate interacting proteins were identified through affinity purification coupled with tandem mass spectrometry experiments, but only several proteins are discussed and focused in Fig 1A-C.

The reviewer is correct in highlighting that numerous interactors were identified but describing all these in detail would necessitate a manuscript in itself. Instead we chose to highlight: proteins that have been previously identified as PEAK1 binding partners (Grb2), as validation of the approach; previously identified PEAK binding partners where selectivity towards PEAK1, PEAK2, and PEAK1/2 was not known (PP1 family members, 14-3-3); and novel binding partners with interesting selectivity (FARP1, SPRY4, and CAMK2 family members). However we appreciate that this strategy is essentially candidate driven. To complement this approach, in the revised manuscript we have undertaken bioinformatic analysis of the interactomes for each of the PEAK complexes in order to identify enriched pathways/processes (**New Supp Table 2**). Of note, this identified enrichment of calcium-related pathways that include CAMK2D/G (specifically postsynaptic signalling for PEAK1 and PEAK2 homotypic complexes and both myometrial and cardiac signalling for all 3 complexes), supporting the subsequent focus on CAMK2/G in the manuscript. These new data are described on **Page 6**.

6. *Why you focus on PEAK1 rather than PEAK2? What's the difference between them? You have demonstrated that CAMK2 interact with both PEAK1 and PEAK2.*

PEAK1-3 are structurally related pseudokinase scaffolds that have contrasting signalling outputs. All three have been implicated in cancer, although the oncogenic role of PEAK1 is more firmly established. This, and the association of PEAK1 with triple negative breast cancer and pancreatic cancer, two malignancies with limited targeted therapeutic options, justified the focus on PEAK1. This is more clearly indicated in the revised manuscript (**Page 3, 7**).

7. *I suggested enhancing the aesthetics of the volcano map displayed in Figure 1. The color is not favorable. Additionally, the font size of the axis labels should be adjusted to ensure they are not too small compared to other text elements. Similar improvements should be made to the other figures as well, as the current presentation is not satisfactory for publication.*

We thank the reviewer for this constructive comment and have improved the presentation of the Volcano plot in **Fig 1**. We have also enhanced the quality of other Figures, including **Supp Figs 5 and 8**.

8. *Most of WB figures in your manuscript do not have reference strips (such as GAPDH, Actin, etc.). Why not use these? Are you regarding 14-3-3 as reference? If so, why you use 14-3-3 as a reference?*

All of the total lysate blots do have loading controls, and we have used β -actin, tubulin or 14-3-3 for this purpose. We would like to highlight that there is no 'perfect' loading control, and even the use of β -actin and tubulin has been questioned at times in the literature, in part due to differences in expression across cell lines/tissues. We have recently been using 14-3-3 as a loading control since total 14-3-3 levels are relatively stable within a given cell or tissue type (Pennington *et al*, *Oncogene* 37, 5587, 2018) and we have experienced no problems in terms of quantification or reproducibility (see for example: Hou *et al*, *Sci Sig* 15, eabj3554, 2022; Yang *et al*, *Oncogene* 42, 833, 2023). Furthermore, 14-3-3 blotting is not saturated within the range of the protein amounts utilized (15-30 micrograms) (Reviewer Figure 3, this rebuttal). To clarify the use of this loading control, appropriate text has been added to the figure legend where it is first used (**Figure 2, Page 41**).

9. *In line 234, the first appearance of KD requires detailed definition.*

This abbreviation has been defined as dissociation constant (**Page 11**).

10. *The resolution of the Extended Figure 5 and 8 needs to be improved.*

This has been undertaken (**Supp Figs 5 and 8** in the revised manuscript).

11. *You explored the effect of knockdown of either CAMK2D or G on the PEAK1-dependent cell migration and invasion. Therefore, in Figure 7B, the p-value of the difference between Vector and PEAK1 in si-CAMK2D or G should be marked.*

This is not significant because of the CAMK2D/G knockdown. We have added 'NS' to the figure to emphasize this (**Revised Fig 7B**).

12. *Since RA306 and gene knock-out have the same effect on the impairment of tumor growth,*

which of these two targets is more likely to be applied in clinic? Is there any clinical research basis on these two targets?

As highlighted in detail in the Discussion (**Page 19**), PEAK1 is challenging to target therapeutically, since it does not bind ATP or GTP, the dimerization interface is relatively large and difficult to target using small molecule drugs, and PEAK1 is an intracellular protein and therefore not accessible by standard therapeutic antibody approaches. Degradation of PEAK1 via a PROTAC represents one potential strategy but requires the identification of a small molecule that selectively binds PEAK1. However, our work now identifies CAMK2 as an actionable target downstream of PEAK1 that can be inhibited using RA306, which was originally developed by Sanofi for treatment of heart disease. Importantly, as we indicate (**Pages 19-20**), RA306 avoids central nervous system side-effects and exhibits good oral bioavailability, highlighting its potential for oncology re-purposing. To our knowledge RA306 is still in pre-clinical development for treatment of heart disease and our study is the first to evaluate its activity in a cancer setting.

Reviewer #4 (co-reviewed with Reviewer #3):

I co-reviewed this manuscript with one of the reviewers who provided the listed reports.

Reviewer #5 (oncogenic kinase signaling):

This reviewer indicated that 'Overall, this is an important study that identifies the mechanism of action of PEAK1 and establishes it as a novel regulator of Ca²⁺ and certain PTKs signaling. In addition, the study offers a new strategy to combat PEAK1-overexpressing triple negative breast cancer.'

The reviewer had some comments that should be addressed to allow publication.

1) Fig. 1 E&F is problematic. In both, the lower panels (CAMK2G) are smeared, and the number of dots seems to be much below the numbers that appear in the quantification. Another point is the lack of signals of CAMK2G/PEAKs in the nucleus, which is unlike the nuclear signals of CAMK2D/PEAKs. Since CAMK2G does localize to the nucleus (at least in some cells) the reason for the lack of PLA signal is not clear and should be explained.

We have provided better quality images for CAMK2G PLA in **Revised Figures 1E and F**. The number of dots were quantified by an established protocol using Image J software. Nuclear dots indicative of close proximity to PEAK1/2 were detected for both CAMK2D and CAMK2G.

2) In Fig. 2, what are the units of the relative p-CAMK2 level. The units are missing in other figures as well.

This is a ratio of pCAMK2/total CAMK2 and is in arbitrary units (au). We have amended the graphs and figure legends throughout the manuscript accordingly.

3) In many figures it is suggested to move the result panel to the top of the figure (e.g. in Fig. 2A and 2C move the second panel (CAMK-pT287) to the top), because these are actually the important parts of the figures.

We have undertaken this for all relevant figures.

4) In Fig. 3D, what are the two bands seen in the flag blot of del 261-300 and 261-324?

These are degradation products. They have been have marked on the gels using an asterisk that is explained in the corresponding figure legend (**Revised Fig 3D, Page 42**).

5) *In Fig. 3E, why is the deletion marked “d” and not del? The MW size of d1-260 seems to be too high.*

We have normalized nomenclature of the deletion mutants across all figures and in the text, using Δ for deletion.

The mobility of the 1-260 deletion mutant is correct.

6) *Fig. 3F is problematic and should be replaced. It seems that the Del 301-324 is taken from another blot, but this is not properly marked or explained. It is important to show the results on one blot for a better side by side comparison. In addition, the band in PEAK1-vector is strange. Please check.*

We apologize for the suboptimal presentation of this figure. All lanes are from the same gel and blot. As indicated in the response to Reviewer 1, a lane was cropped from the blot since this corresponded to an irrelevant mutant. We have indicated the cropping by a dotted line in the **Revised Fig 3F** and explained this in the figure legend (**Page 42**). The full blot is shown as an uncropped image in source data and in **Reviewer Figure 1 (above)**.

The band in PEAK1 vector is endogenous PEAK1.

7) *In the extended data figure 2A, there is something wrong in the top legends.*

We have improved the presentation of this figure.

7) *In extended data figure 4, why is the deletion marked Δ and not del or d as in other figures?*

As indicated above, we have normalized nomenclature of the deletion mutants across all figures.

8) *The term SFK is not spelled out anywhere in the text. Can the authors provide information on the relevant Src family kinases involved?*

We apologize for this oversight and have defined SFK at first mention in the text of the revised manuscript (**Page 8**).

In terms of the SFKs responsible, triple negative breast cancer cells express several SFKs including Src, Yes, Fyn and Lyn, but expression of the latter is particularly high, and since Lyn also regulates PEAK1 tyrosine phosphorylation, it represents an interesting candidate for further investigation (Croucher *et al*, Cancer Res 73, 1969, 2013; Hochgrafe *et al*, Cancer Res 70, 9391, 2010; Sausgruber *et al*, Oncogene 34, 2272, 2015). We have provided additional text clarifying this on **Page 16**.

9) *In several figures, the authors normalized the tested parameters by dividing the intensity of the tested parameter by the intensity of the total band. This is problematic because the different antibodies may have a different recognition linearity (namely, the change in intensity by each antibody does not necessarily reflects the difference in the amount of proteins /*

phosphorylation). It is suggested to use a calibration curve for each of the antibodies to correlate the intensity to the protein amounts.

Reviewer Figure 3. Western blotting of lysate dilution series.

Normalizing the phospho signal for total is a standard approach used widely in the literature (see for example our previous papers: Ma *et al*, Nature Comms 10, 296, 2019; Rodgers *et al*, Nature Comms 12, 3140, 2021). It is of course important to confirm that the signals obtained for the respective antibodies are not saturated, and to address the reviewer's concern, we have confirmed that the Western blot signal for phospho and total CAMK2 and PLCγ1, key antibodies used in the study, are not saturated for the protein amounts utilized (15-30 micrograms) (**Reviewer Figure 3, this rebuttal**). We believe that this is a justified approach for determining whether there is a change in, for example, the relative phosphorylation of CAMK2. We totally accept that the approach is only semi-quantitative, but here it is important to appreciate that as indicated in an informative review '*even when the ideal housekeeping gene was loaded at a concentration in the linear range, or when the total protein concentration was measured at maximum accuracy to avoid housekeeping proteins altogether, ECL detection of Western blot signals still provides only semiquantitative expression data due to the nonlinearity of the cumulative luminescence and limited quantitative reproducibility*' (Gorr and Vogel, Proteomics Clin. Appl., 9, 396–405, 2015). Consequently, we do not believe that repeating the entire study with the inclusion of calibration curves would add value. Furthermore, our normalization also takes into account a loading control on the same gel in order to correct for technical issues eg in sample loading and/or transfer, and running separate gels with different amounts of protein on them prevents this strategy.

We have added text to the Methods (**Page 23**) indicating that in order to ensure that phospho and total antibodies were used at non-limiting concentrations, Western blot analyses were undertaken on dilution series of cell lysates.

10) Although this point does not affect the results, in my mind the addition of 12.5 mM beta glycerophosphate to RIPA is not necessary, as the other components of the buffer are sufficient to inhibit phosphatases.

We thank the reviewer for this advice.

1) We have responded to this comment from Reviewer #1 -

Reviewer #1 (Remarks to the Author):

The authors have adequately answered all issues raised. There is only one comment arising for consideration.

Based on the study, high expression of PEAK1 would enhance Ca²⁺ level. Once PEAK1 is overexpressed, it elevates intracellular Ca²⁺ immediately, activating Ca²⁺ signalling. Prolong activation of Ca²⁺ signalling is apoptotic. The revised Fig2G, H, only show single time point experimental result. We cannot tell from this whether there was a negative feedback mechanism to reduce Ca²⁺ in the later time point. The original manuscript, Figure 2G and H did indeed present sequential data showing this phenomenon. The authors might consider including the sequential data in the supplementary figures to exclude the possibility that the PEAK1 interactome could be involved into anti-apoptotic pathways.

We have improved the methodology for our Ca²⁺ assays and used Fluo-4 AM in the revised manuscript rather than Fluo-2 AM that was used in the original submission. The assay results indicate that there is a significantly increased level of intracellular Ca²⁺ in the PEAK1-overexpressing cells at steady-state (Fig 2G-H). We highlight this in the Discussion and acknowledge the link between enhanced Ca²⁺ and apoptosis. However we believe that PEAK1 likely provides pro-survival signals to overcome the potential Ca²⁺-induced apoptosis, for example via Grb2, a member of the PEAK1 interactome. Consequently we have added the following statement in the Discussion (Line 387):

' We note that sustained elevation of intracellular Ca²⁺ can be pro-apoptotic²⁰, but this is likely countered by survival signals emanating from PEAK1, for example via Grb2/Ras/Erk⁵. '

2) We have improved the data presentation in Figure 7 regarding the frequency of metastasis. We have moved the plot of luciferase intensities of tumour metastasis to lung for each treatment group to Supp Fig 11E and provided a new plot and statistical analysis in Fig 7F that highlights the absence of significant metastatic burden in the 3 treatment groups. The interpretation of the data is the same, but the presentation is now clearer.

3) We have revised the manuscript text and all figures to match formatting requirements, in particular we have provided actual p values on each figure.

We hope that you now find the manuscript suitable for publication,

Yours sincerely,